# Efficacious human metapneumovirus vaccine based on AI-guided engineering of a closed prefusion trimer

Mark J. G. Bakkers [1,3], Tina Ritschel [1,4], Machteld Tiemessen[1], Jacobus Dijkman [1,5,6], Angelo A. Zuffianò[1,7], Xiaodi Yu [2], Daan van Overveld [1], Lam Le[1], Richard Voorzaat [1], Marlies M. van Haaren[1], Martijn de Man[1], Sem Tamara [1], Leslie van der Fits[1], Roland Zahn [1], Jarek Juraszek[1,8] & Johannes P. M. Langedijk [1,3,8] ✉

The prefusion conformation of human metapneumovirus fusion protein (hMPV Pre-F) is critical for eliciting the most potent neutralizing antibodies and is the preferred immunogen for an efficacious vaccine against hMPV respiratory infections. Here we show that an additional cleavage event in the F protein allows closure and correct folding of the trimer. We therefore engineered the F protein to undergo double cleavage, which enabled screening for Pre-F stabilizing substitutions at the natively folded protomer interfaces. To identify these substitutions, we developed an AI convolutional classifier that successfully predicts complex polar interactions often overlooked by physics-based methods and visual inspection. The combination of additional processing, stabilization of interface regions and stabilization of the membrane-proximal stem, resulted in a Pre-F protein vaccine candidate without the need for a heterologous trimerization domain that exhibited high expression yields and thermostability. Cryo-EM analysis shows the complete ectodomain structure, including the stem, and a specific interaction of the newly identified cleaved C-terminus with the adjacent protomer. Importantly, the protein induces high and cross-neutralizing antibody responses resulting in near complete protection against hMPV challenge in cotton rats, making the highly stable, double-cleaved hMPV Pre-F trimer an attractive vaccine candidate.

Human metapneumovirus (hMPV) is an enveloped, negative-sense RNA virus belonging to the family Pneumoviridae[1]. hMPV is a major cause of both upper and lower respiratory tract infections in infants, young children, the elderly and among immunocompromised persons or those with underlying chronic medical conditions[2-12].

The hMPV fusion protein F is a class I fusion protein present on the viral surface and essential for viral entry. It is initially synthesized as a single chain precursor F0, which is cleaved by host cell proteases into disulfide-linked F2 and F1 subunits, with the latter bearing the fusion peptide at its N-terminus. F mediates the fusion of viral and host cell membranes through a large conformational change from

[1]Janssen Vaccines & Prevention BV, Leiden, The Netherlands. [2]Structural & Protein Science, Janssen Research and Development, Spring House, PA 19044, USA. [3]Present address: ForgeBio B.V., Amsterdam, The Netherlands. [4]Present address: J&J Innovative Medicine Technology, R&D, New Brunswick, NJ, USA. [5]Present address: Van 't Hoff Institute for Molecular Sciences, University of Amsterdam, Amsterdam, The Netherlands. [6]Present address: Amsterdam Machine Learning Lab, Informatics Institute, University of Amsterdam, Amsterdam, The Netherlands. [7]Present address: Promaton BV, Amsterdam, The Netherlands. [8]These authors contributed equally: Jarek Juraszek, Johannes P. M. Langedijk. ✉e-mail: hlangedijk@forge-bio.com

the metastable prefusion (Pre-F) to the highly stable postfusion (Post-F) conformation[13–15]. Since Pre-F is the target for neutralizing antibodies, development of an efficacious hMPV vaccine is aimed at stabilizing this conformation[15–18]. The overall fold of the hMPV F trimer closely resembles that of the respiratory syncytial virus (RSV) F protein, the other known human pneumovirus[19]. However, a notable distinction lies in their proteolytic activation: hMPV F0 is cleaved at a single monobasic cleavage site by a host cell protease (e.g. TMPRSS2), whereas RSV F0 undergoes cleavage at two separate polybasic cleavage sites (positions 109 and 136) by a furin-like protease, resulting in the releases of a 27-amino acid fragment (p27) that is not part of the mature protein[20]. Structural studies have shown that for both pneumovirus F proteins, the fusion peptides located at the N-terminus of F1 are buried within the trimer's central cavity. For RSV, the C-terminal region of F2 is also positioned inside this cavity. In contrast, the hMPV F trimer lacks structural evidence to suggest that its F2 C-terminus occupies the cavity and the longer F2 and fusion peptide imply a potential obstruction in the trimer cavity, possibly hindering trimerization. An additional cleavage event, N-terminal to the primary cleavage site that truncates F2, akin to RSV F's double processing which releases the p27 domain, could resolve this obstruction.

For efficacious vaccines, stabilizing the prefusion conformation of viral fusion proteins is critical[21–25]. Our data indicate that complete closure of the trimer, akin to the RSV F trimer's structure, requires the additional cleavage. We introduced several stabilizing substitutions into hMPV F, guided by our AI model. The model, named ReCaP (Residue Classification for Protein Design), is a protein structure-based, residual convolutional neural network that was used to identify and optimize positions exhibiting highly confident amino-acid misclassifications. Compared to generative models or physics-based methods, the model is particularly successful in optimizing charged and solvent-mediated interactions. Here, we show that the resulting hMPV F trimers – efficiently cleaved at both positions, trimerized without foldon and locked in the prefusion closed conformation – elicit potent neutralizing antibody responses in mice and provide protection in both upper and lower respiratory tract of immunized cotton rats challenged with hMPV.

## Results

### Analysis of hMPV F processing

hMPV neutralization assays typically utilize immortalized cells and require exogenous trypsin to facilitate the cleavage of F0 at its monobasic cleavage site for efficient infection[1,26–28]. To examine if TMPRSS2 could also enable this cleavage, we transfected HEK293 cells with hMPV F0 plasmid alone or in combination with plasmids encoding human TMPRSS2 or furin to evaluate their effect on processing. Western blot (WB) analysis revealed incomplete processing of F0 in HEK293 cells. Co-transfection with furin reduced F0 processing whereas co-transfection with TMPRSS2 enhanced it, albeit not to completion (Fig. 1A). In contrast, WB analysis of hMPV harvested from fully differentiated human airway epithelial cells (hAECs) infected with hMPV showed complete processing into F1 and F2 fragments, suggesting that these primary cells can facilitate complete F0 cleavage in egressing hMPV particles without co-transfection of additional enzymes (Fig. 1B, C).

In contrast to RSV F, where the F2 C-terminus is located in the central cavity of the trimer, for the hMPV F trimer, the last 11–15 C-terminal residues of F2 remain unresolved[15,29,30]. The last visible residue is positioned on the trimer's exterior, without structural indication that the peptide chain traverses into the trimer cavity. The hMPV F fusion peptide is extended by four residues compared to the RSV fusion peptide, occupying a larger portion of the cavity. Moreover, considering the proteolytic trimming of the F2 subunit's C-terminal arginines by ubiquitous exoproteases, the remnant hMPV F2 C-terminal region would be six residues longer than its RSV counterpart (Supplementary Fig. 1). This implies that hMPV F's trimer cavity would have to accommodate thirty more residues than RSV F's – a special constraint suggesting the extra unresolved F2 residues might extend outward from each protomer in a disordered fashion. An extension outward is actually observed for several monomeric hMPV F structures that are not able to trimerize due to the extension of the F2 helix up to residue P98 which causes clashes with the adjacent protomer[17,31]. Alternatively, the obstruction by the long F2 in the monomers and the absence of structural data for the F2 C-terminal region for closed trimers might suggest a secondary cleavage event, reminiscent of the RSV F's p27 cleavage.

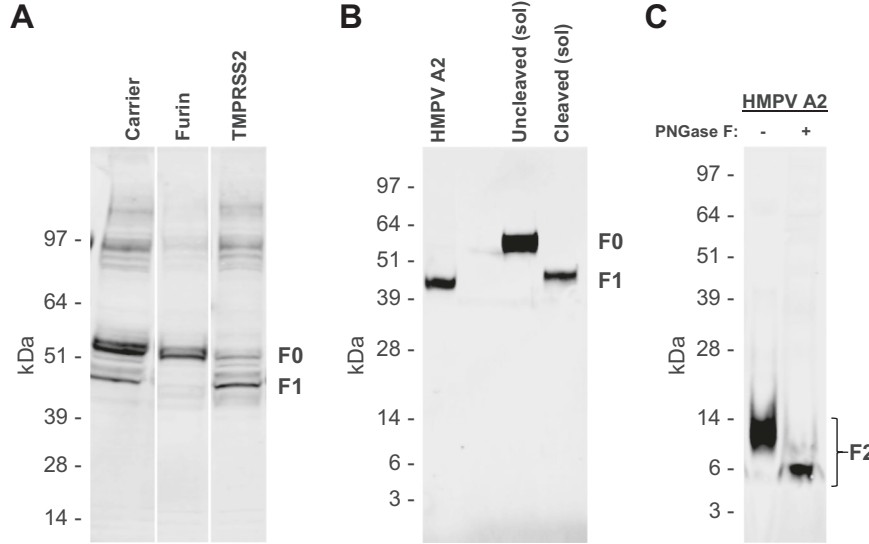

**Fig. 1 | Analysis of hMPV F cleavage. A** WB after reduced SDS-PAGE of full-length hMPV F. Lanes contain cell lysate of HEK293 cells transfected with hMPV F or with hMPV F co-transfected with 20% furin or 20% TMPRSS2. **B** WB using F1-specific and **C** F2-specific sera with supernatants from hMPV infected hAECs, 4 days after infection. F1-specific blot shows soluble cleaved and uncleaved hMPV F as reference. F2-specific blot shows PNGase and untreated lanes. Experiments of **A**–**C** were performed twice, with similar results, a representative experiment is shown.

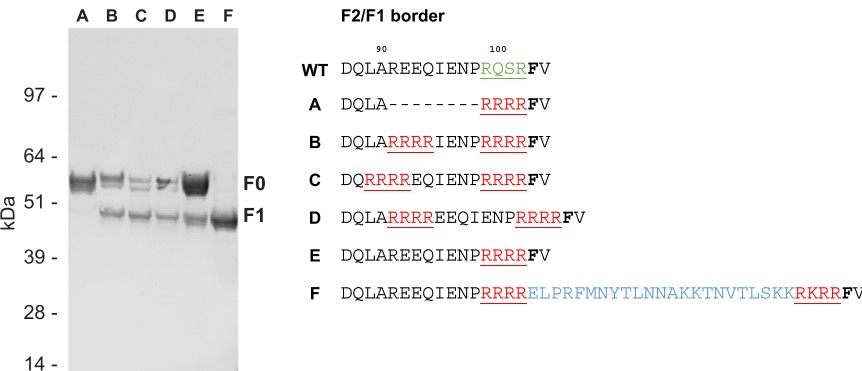

**Fig. 2 | Optimization of hMPV F cleavage.** WB after reduced SDS-PAGE of cell culture supernatant of Expi293F cells transfected with indicated hMPV F constructs. F1-specific WB shows uncleaved (F0) and cleaved (F1) hMPV F. The table on the right highlights the different cleavage site designs (green: TMPRSS2 cleavage site, red: furin cleavage site, blue: RSV's p27 peptide). The phenylalanine at the start of the fusion peptide is highlighted in bold. The experiment was performed twice, with similar results, a representative experiment is shown.

To investigate the possibility for additional cleavage events, the supernatant harvested from hMPV infected hAECs, was separated on a reduced SDS-PAGE gel and excised lower molecular weight bands were digested with Lys-C (Supplementary Fig. 1). After digestion, the peptides were eluted from each excised band and analyzed by liquid chromatography-mass spectrometry (LC-MS/MS) to determine the identity of the C-terminal F2 peptide variants. Despite the high protein background, the F2 C-terminal peptides could be detected in the excised bands. A peptide representative for a single cleavage event after the monobasic cleavage site at position 102 with an additional cleavage of the C-terminal Arg[102] by an endoprotease was the most abundant peptide detected. However, a significant portion of the peptides were C-terminally truncated. These data confirm that the F protein of hMPV that egresses from fully differentiated bronchial hAECs are cleaved and truncated but a dominant second cleavage site could not be detected (Supplementary Fig. 1).

### Cleavage of hMPV F with F2 truncation

In an effort to establish processing of recombinant soluble hMPV F in a non-native environment such as Expi293F cells, we engineered a furin cleavage site that would allow native-like processing. We designed multiple hMPV F variants, including a variant that contained the p27 domain of RSV F which is known to be cleaved very efficiently by furin at two sites[32]. WB analysis revealed that neither the introduction of a single nor a double furin cleavage site achieved complete cleavage of hMPV F. It was only with the integration of the RSV F p27 domain that full cleavage of the hMPV F protein was obtained (Fig. 2).

Since F2 could obstruct F trimerization and the MS analysis of egressed hMPV F did not yet confirm a second cleavage site, the structural impact of F2 truncation was investigated in recombinant proteins. In order to measure impact on trimerization we did not fuse the F protein to a typical trimerization domain but instead stabilized the second heptad repeat (HR2) in the stem region by introducing bulky hydrophobic amino acids at positions 473, 477 or 484 in the stem region. Because the triple helix HR2 stem does not exhibit a classical coiled-coil structure until residue 473, but instead forms a structure akin to an "inverted pyramid" with empty space inside, we opted for large hydrophobic residues at positions 473 and 477 to facilitate trimerization by optimizing amino acid side chain packing. Additionally, to potentially increase solubility and stability, four charged residues from the RSV F stem were introduced at positions 475, 476, 478, and 479, where 475 and 479 could form a stabilizing ionic interaction.

hMPV F ectodomain variants of subtype A2 were transfected at 96-well scale in Expi293F cells and harvested after three days. To evaluate the impact of F2 truncation and HR2 stabilization on

trimerization, supernatants of transfected cells were analyzed for F trimer expression using analytical size exclusion chromatography (SEC) (Supplementary Fig. 2a). Additionally, the F conformation was assessed by Biolayer Interferometry (BLI) using Octet with a prefusion-specific monoclonal antibody (mAb) ADI-61026[29], a potent neutralizing antibody that targets site Ø at the apex of the trimer, mAb DS7 which targets a Post-Fusion or a Pre-F-to-Post-F intermediate conformation[30,33], and a prefusion-specific mAb MPV458[31], which binds to both Pre-F monomers and splayed-open trimers since the epitope is buried upon closure of the apex (Supplementary Fig. 2b). MPV458 showed very low neutralizing activity compared with the site Ø – specific mAb ADI-61026 as shown in a neutralization assay using the hMPV CAN97-83-GFP strain in combination with fully differentiated hAECs which recapitulate human lung physiology and are considered an accurate model to study virus neutralization[34] (Supplementary Fig. 3, Supplementary Methods).

Trimer expression could be detected by SEC only when F was cleaved, F2-truncated and the HR2 stem region stabilized. Stabilizing the positions 473, 477 and 484, or incorporating additional substitutions based on the stem region of RSV F, resulted in a notable increase in trimeric F formation, as well as decrease in monomer species based on SEC analysis (Fig. 3A, Supplementary Fig. 2a). This shift from monomeric to trimeric species was consistent with the antigenicity profiles of the F variants - those containing a higher proportion of trimeric species, as indicated by SEC, showed a stronger signal for the site Ø - specific ADI-61026 and a weaker signal for the trimer interface-, or monomer – specific MPV458 (Fig. 3A, Supplementary Fig. 2b). This indicates that hMPV F can form a closed trimeric conformation when it is cleaved and F2 is truncated. In contrast, the cleaved variant with stabilized HR2 stem, but without F2 truncation, exhibited significant binding to MPV458, indicating the presence of open conformations or monomeric species.

To verify the optimal F2 truncation, a series of hMPV F variants were designed with systematic truncations at the F2 C-terminus. Each variant incorporated HR2 stem stabilization (VII) (Fig. 3A, Supplementary Fig. 2c), the E453Q substitution, and four arginines preceding the RSV p27 domain (Fig. 3B). hMPV-p27 variants were transfected in Expi293F cells and cell culture supernatants were tested on SEC for trimer expression and by BLI to evaluate binding to mAbs ADI-61026, MPV458 and DS7. Variants with more extensive F2 truncations, while having somewhat lower expression, exhibited a prominent trimer peak in SEC at 4.46 min and bound more strongly to the Pre-F-specific mAb ADI-61026 as opposed to MPV458 (Fig. 3B, Supplementary Fig. 2d). This binding profile is indicative of a more closed trimer conformation in comparison to variants with longer F2 C-termini. Notably, the variant

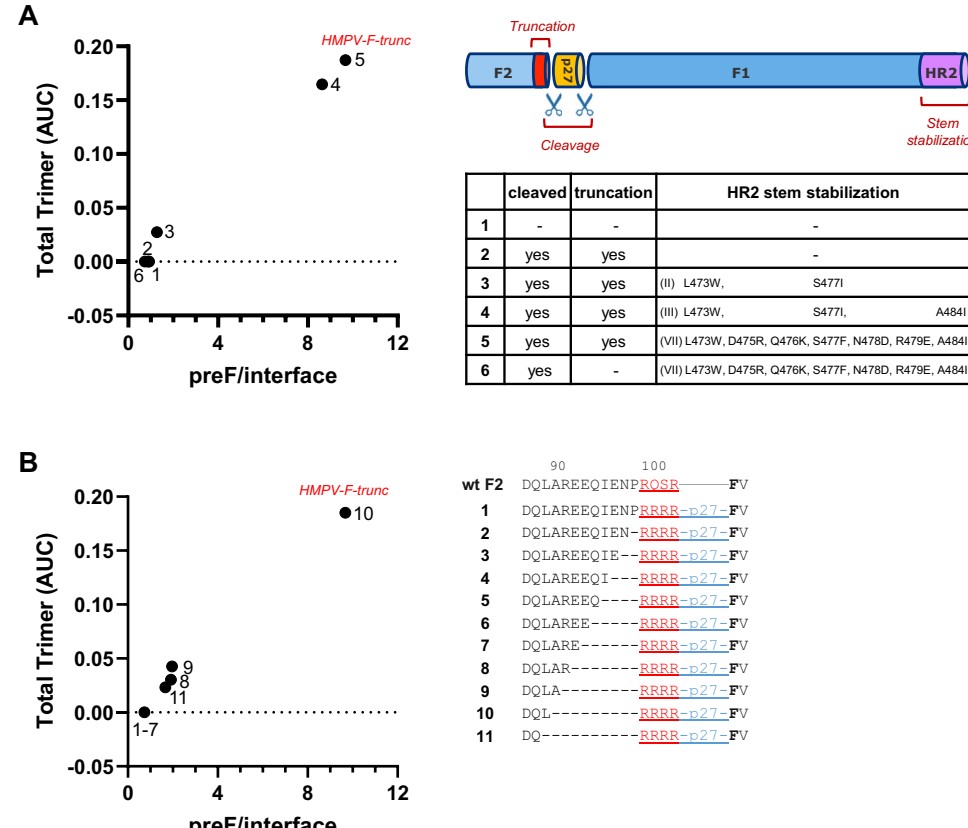

**Fig. 3 | Characterization of hMPV F trimerization. A** Scatter plot of trimer expression versus trimer:monomer ratio. hMPV F trimer expression using analytical SEC and BLI using Pre-F-specific ADI-61026 and monomer-binding MPV458 of cell culture supernatant of Expi293F cells expressing hMPV F (Supplementary Fig. 2a, b). Trimer expression level against ADI-61026:MPV458 binding ratio is shown for the constructs numbered in the table. (top right) A schematic representation of the soluble F protein highlighting the different design elements. All constructs contained E453Q that neutralizes the ring of negative charge repulsion at the three-fold axis above the stem[17,69], constructs 2-6 were double cleaved using RSV p27 and 2–5 have an F2 truncation with four arginines C-terminal of residue L89 ($_{99}$RRRR$_{102}$). **B** Plot of trimer expression level against ADI-61026:MPV458 binding ratio is shown for hMPV F variants with truncated F2 domains as indicated in the Table on the right. hMPV variant with optimal F2 truncation is labeled hMPV-F-trunc, which is similar as construct 5 in (**A**). The table highlights the different truncations (red: furin cleavage site, blue: RSV's p27 peptide).

designated as hMPV-F-trunc, which contains a furin cleavage site at position 90−93, showed not only the highest trimer peak in SEC analysis, but also superior antigenicity profile, characterized by the highest ratio of ADI-61026 / MPV458 binding and no binding to DS7 (Fig. 3B, Supplementary Fig. 2c, d).

## Stabilizing the prefusion conformation

Two distinct methods were employed to predict substitutions for improved stability of the prefusion conformation of the hMPV F trimer (Fig. 4A). The first method utilized the physics-based fixed-backbone or coupled moves design algorithms[35,36] implemented in Rosetta (https://www.rosettacommons.org/). The algorithm is a computational method used to design protein interfaces and utilizes a Monte Carlo simulation technique that iteratively optimizes the interface by making correlated changes to the side chains and backbone angles of interacting proteins. This approach helps in accurately modeling the complex movements and structural adjustments that occur during protein interactions, leading to more reliable predictions and designs. The second method involved a physics-agnostic approach that we developed based on an artificial intelligence (AI) classifier. The classifier, built as a residual convolutional neural network (Fig. 4A) and designed to predict masked amino acids based on their voxelized atomic microenvironments, was trained on a large dataset of protein structures from the Protein Data Bank (PDB), effectively enabling it to recognize typical, stable and relatively well expressing protein structure. In our attempt, we decided to limit the information available to the model to the minimum by only using the 4 atomic channels corresponding to the positions of carbon, oxygen, nitrogen, and sulfur. We did not include any precalculated features such as partial charges, solvent accessibility or positions of hydrogen atoms. We used residual connections to facilitate information flow through the network during training, and learn a richer representation specific to the classification problem. For details of model training and inference see "Methods" section. Our approach resulted in an average accuracy of 65% for the wild-type amino-acid prediction task (Supplementary Fig. 4). We evaluated various models to optimize voxelization resolution (see "Methods" section for more details about voxelization procedure), discovering that reductions below the selected threshold of 0.8 Å did not enhance wild-type prediction accuracy. Additionally, attempts to improve performance by incorporating atomic charges and adding more convolutional layers were unsuccessful. For the purpose of identifying stabilizing substitutions, we used all 20 amino-acid probabilities from the model's output (See Fig. 4A). When we applied the classifier to the hMPV prefusion F structure (PDB ID 5WB0), we focused on identifying amino acid sites with suboptimal or unfavorable atomic microenvironments, particularly at or near protomer-protomer interfaces. To pinpoint sites for substitution, we specially searched for significant discrepancies in the classifier's predictions, where the amino acids suggested by our model differed markedly in biochemical properties from the corresponding wild-type residues.

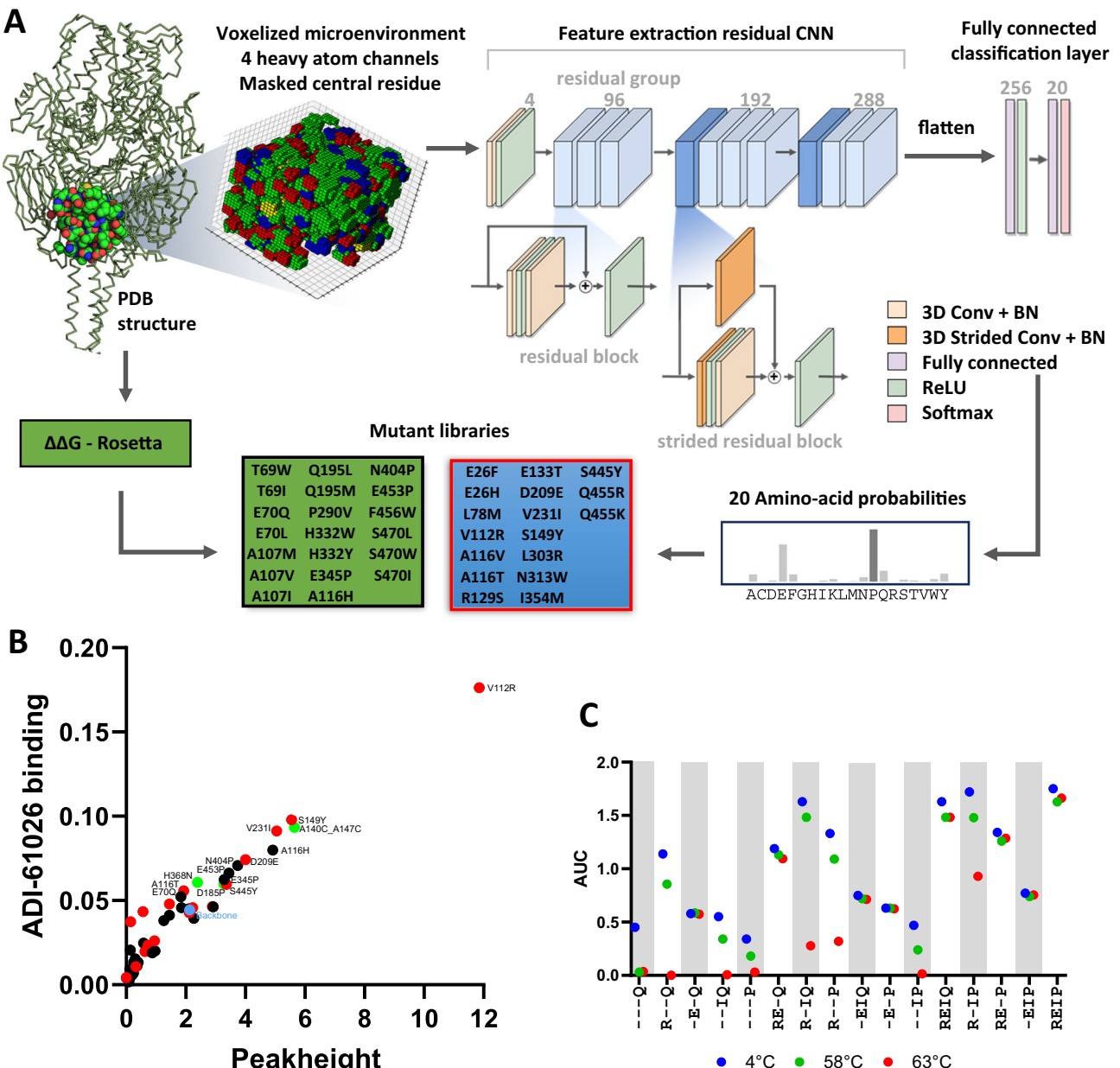

**Fig. 4 | Identification of prefusion F stabilizing substitutions. A** Overview of the stabilizing mutation computational screening workflow, including the ReCaP model alongside Rosetta's coupled moves and fixed-backbone design methods. In the ReCap model, the softmax probabilities determine the stability of micro-environments, with mutations exhibiting high probabilities being chosen for screening if the environment is deemed unstable. The neural network architecture includes a feature extraction component, constructed from convolutional layers equipped with skip connections and the ReLU activation function. Batch normalization (BN) is implemented in the second residual block to enhance stability and performance. The network ends with a fully connected classification layer, followed by a softmax function that normalizes the output into probabilities across the 20 classes. For the Rosetta part of the workflow, mutations are selected for screening based on having the lowest calculated $\Delta \Delta$G values. The libraries designed with Rosetta and the AI model are listed in black and red frames respectively. **B** hMPV F trimer expression using analytical SEC and Pre-F-specific ADI-61206 binding of cell culture supernatant of Expi293F cells expressing hMPV F. Plot of trimer expression level against ADI-61206 binding for each mutant is shown. Markers are black for Rosetta, red for AI-predicted substitutions and green for previously published stabilizing substitutions[15,17,37,38]. The backbone is similar to hMPV-F-trunc with additional H368N substitution and is indicated in blue. **C** Trimer expression levels for combinations of stabilizing substitutions V112R, D209E, V231I, and E453Q/P. Systematic combination of substitutions at these four positions are shown on the x-axis and markers show total trimer after storage at 4 °C (blue), or after heat treatment for 30 min at 58 °C (green) and 63 °C (red).

Supernatants of Expi293F cells transfected with plasmids encoding variants of hMPV-F-trunc with single amino acid substitutions based on Rosetta predictions or AI predictions and several of the most successful published stabilizing substitutions were assessed for trimer content using SEC and prefusion F content using octet measurement with ADI-61026 (Fig. 4B, Supplementary Fig. 5). Similar as for the previous screens, a correlation was observed for trimer content and

binding to ADI-61026. Substantially higher trimer expression increase was observed for substitutions derived from the AI/ML predictions compared to other methods. Especially V112R, located at the helical kink in the fusion peptide showed a very strong increase in trimer expression compared with all other substitutions including the successful 140–147 disulfide bond previously described[16,17,37] (Fig. 4B). The cavity-filling substitution S149Y improves the interaction between the

sheet and the 131–140 helix. Three stabilizing proline substitutions were identified in the lower part of the fusion protein, closer to the stem. N404P, E453P, and E345P introduce rigidity in the flexible loop regions and could simultaneously facilitate the folding process and hinder refolding. Additionally, E453P rigidifies the flexible HR2 loop and neutralizes the ring of negative charge repulsion at the three-fold axis above the stem. Another example of stabilization in the stem region is the S445Y substitution, which acts by filling a shallow pocket on the interface with the HR2 and interacts with the F464. Next, the impact on expression and stability was tested for selected stabilizing substitutions V112R, D209E, and E453P that stabilize the protomer interface and the cavity-filling V231I substitution that was predicted by the AI model and was also described previously[17]. All constructs were double cleaved using RSV p27, contained the HR2 (VII) optimization, the F2 truncation, to stabilize trimer formation and the H368N[38] to increase expression levels. The single substitutions confirmed the strong impact of V112R on expression levels (Fig. 4C). Substitution of E453P instead of E453Q showed slightly lower expression but the E453P substitution increased stability at 58 °C. For all combinations the conservative substitution D209E showed a very strong stabilizing effect relative to the parental construct and even with a single D209E substitution the trimer remained stable after 30 min of heat treatment at 63 °C. Combination of all four stabilizing substitutions resulted in the most stable F trimer with the highest expression (Fig. 4C). Next, uncleaved, single cleaved and double-cleaved variants were tested to establish the additive effect of cleavage, double cleavage and the stabilizing substitutions on trimer stability. Despite the stabilizing substitutions, the uncleaved Pre-F was expressed as a monomer, the cleaved variant was mostly trimeric but still showed some monomer expression and the double cleaved design with the F2 C-terminal truncation was fully trimeric although not fully cleaved based on the sharpening of the trimer peak after additional furin addition (Supplementary Fig. 6). For the cleaved variants with stabilizing substitutions in the trimer interface, the double cleaved variant showed slightly lower binding with the anti-interface – specific mAb MPV458 (Supplementary Fig. 6).

## Stabilized double-cleaved hMPV F retains a closed trimeric prefusion conformation

To evaluate the impact of double cleavage on the Pre-F trimer, fully cleaved stabilized closed Pre-F trimers and a non-cleaved open hMPV Pre-F variant with R102Q, V112R, S149Y, V231I, H368N, and E435P that was fused with a C-terminal foldon (MPV-foldon) and exposes trimer interface epitopes were purified and analyzed (Fig. 5). Two different double cleaved hMPV Pre-F trimers with F2 truncation and HR2 (VII) stabilization were purified: a Pre-F variant with V112R, D209E, V231I, H368N, and E453P which is labeled MPV-2c and a variant with two additional stabilizing substitutions (S149Y and N404P) in different regions labeled MPV-2c2. Full cleavage for the latter two was confirmed by SDS-PAGE, and SEC-MALS showed a symmetric peak at the expected retention time with the correct molecular weight for all three trimers (Fig. 5A, B). Freeze/thaw stability was higher for the closed trimers (Supplementary Fig. 7) and DSF indicated a $Tm_{50}$ corresponding to the main melting event of 67.1 °C for the open non-cleaved trimer, 71.8 °C for MPV-2c and 73.9 °C for MPV-2c2 with two additional stabilizing substitutions, which makes it more thermostable compared to the disulfide stabilized Pre-F trimers (Fig. 5C)[17,18]. The conformation of the proteins was evaluated by BLI using mAb DS7, Pan-F-binding mAbs ADI-18992 and ADI-14448[39], the anti-interface mAb MPV458 and the Pre-F-specific mAb ADI-61026 (Fig. 5D). In contrast to the cleaved Pre-F variants, the foldon-trimerized, non-cleaved Pre-F showed binding to MPV458 indicating a splayed-open trimer and some binding to DS7, indicating that the base of the head domain in the vicinity of HR2 may not be completely in the prefusion conformation. To show the general applicability of the stabilization

strategy for other hMPV strains, the MPV-2c design was applied to the B2 strain and avian MPV. The wt hMPV F B2 showed only monomer expression and low stability whereas the stabilized variant showed high expression level of trimers and a melting temperature of 72.7 °C which was even slightly higher than the strain A1 variant (Fig. 5E, F). BLI using the Post-F or Pre-to-Post-F intermediate specific DS7, the trimer interface-specific MPV458 and the site Ø specific ADI-61026 showed that the stabilized hMPV F B2 trimer was in the prefusion conformation and fully closed in contrast to the wt B2 trimer that showed binding to DS7 and MPV458 (Fig. 5G). Also, the wt avian MPV F showed only monomers and low expression levels whereas the stabilized variant showed a very strong increase of F expression which was mostly trimeric (Fig. 5H).

## Cryo-EM structure of stabilized hMPV Pre-F

To better understand the roles of cleavage, F2 truncation and substitutions in stabilizing the hMPV Pre-F protein, we determined the structures of double-cleaved Pre-F trimers using single-particle Cryo-EM. For a detailed look at the C-terminus of F2 we focused on two variants, MPV-2c with the additional furin site as described above ($_{90}$RRRR) and MPV-2cREKR with an alternative furin site ($_{91}$REKR) with higher similarity to the native sequence. The purified hMPV Pre-F protein was directly applied to cryo-EM grids. Raw EM images revealed no aggregated particles, and 2D classification confirmed the presence of only the trimer form with no monomer form observed. A stable class of Pre-F trimer particles was isolated and refined by imposing C3 symmetry. These structures were resolved to 3.3 Å and 3.1 Å respectively (Supplementary Table 1, Supplementary Figs. 8 and 9) and showed high similarity, with a root-mean-square deviation (RMSD) of ~ 0.55 Å. When compared to the most complete hMPV F structure available (PDB ID: 5WB0[15]), the overall RMSDs were 1.35 Å and 0.99 Å, respectively. Our structures showed the long HR2 stem up to residue Gly487, making them the most complete hMPV structures to date (Fig. 6A, Supplementary Fig. 10a). Substitutions L473W and S477F in HR2 introduced a compact aromatic cluster at the juncture of the three helices (Fig. 6B). Within this cluster, the phenylalanine residues engage in interprotomeric T-shaped aromatic interactions with the tryptophans. Moreover, Q476K covers their exposed phenyl rings, and forms pi-cation interaction with the tryptophans' indole rings contributing to the overall stability of the structure.

Consistent with previous structures, the fusion peptide engages with the HR2 and domain II beta-sandwich of the adjacent protomer[15]. Interactions of the fusion peptide with the adjacent protomer are mediated primarily by F103 and I108 but these interactions show a rather non-optimal side chain packing, with small cavities in between the side chains (Fig. 6D). Interestingly, two stabilizing substitutions A116T and A116H were also confirmed in this protomer interface region (Fig. 4B). The V112R substitution introduces new interprotomeric van der Waals contacts with the side chains of M372 and A373 of the adjacent protomer and an intraprotomeric hydrogen bond with I108's backbone carbonyl. This substitution stabilizes the energetically unfavorable helical kink near the fusion peptide's N-terminus and preserves important interactions that anchor the fusion peptide at the trimer interface (Fig. 6D). The stabilizing D209E substitution is also located at the junction between two protomers and forms intraprotomeric contacts with R205 and the interprotomeric contacts with R252. The wild-type D209 does not form a salt bridge or hydrogen bond with either arginine. In the wild-type structure, a cavity is situated between the D209 carboxylate, the R205 guanidine group, and the backbone of L219. This site likely accommodates a water molecule that mediates interactions with D209's carboxyl. The D209E substitution fills this space more effectively, maintains the local charge balance, forms a hydrogen bond with L219's NH group and also establishes polar contacts with the guanidine group of R205 (Fig. 6C). Interestingly, in the post-fusion conformation, D209 does form an

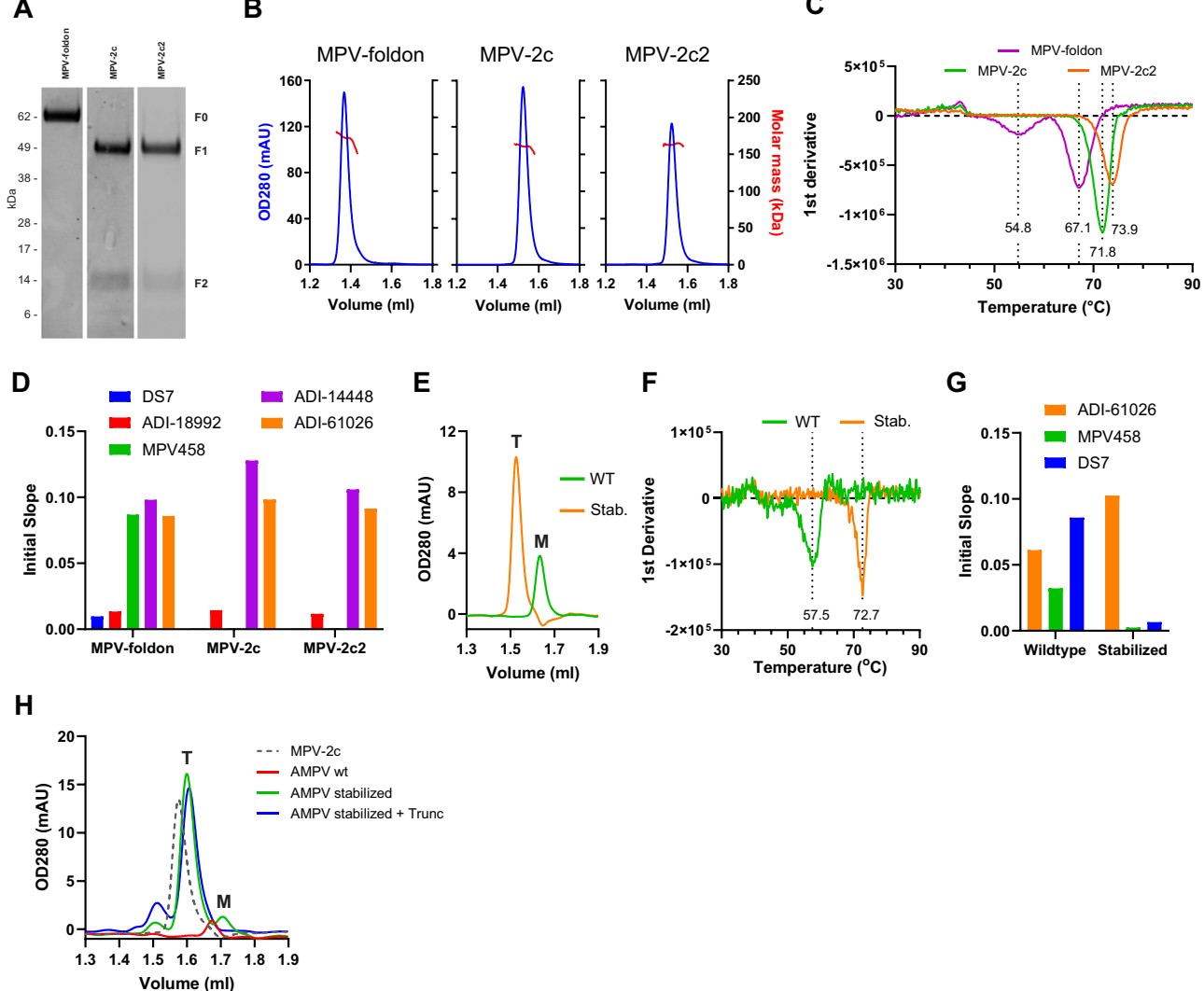

**Fig. 5 | Description and analysis of hMPV Pre-F immunogens.** Analysis of foldon-trimerized non-cleaved hMPV Pre-F (hMPV-foldon) with an open conformation and closed trimers that were fully cleaved, F2-truncated and contained the HR2 (VII) stabilization. MPV-2c contained V112R, D209E, V231I, H368N and E453P substitutions. MPV-2c2 additionally contained T149Y and N404P. **A** Reduced SDS-PAGE, **B** SEC-MALS, and **C** melting temperature using DSF. Experiments of A-C were performed twice, with similar results, representative experiments are shown. **D** The conformation of the proteins was evaluated by BLI using mAb DS7 and prefusion-specific mAbs ADI-18992, MPV458, ADI-14448 and ADI-61026. MPV-2c design elements were transferred to strain B2. **E** SEC profiles showing expression level and quaternary structure of wt and stabilized hMPV F B2. **F** Melting temperature of wt and stabilized hMPV F B2. **G** Antibody binding using BLI of wild type and stabilized hMPV B2 F trimers using mAb DS7 and prefusion-specific mAbs MPV458 and ADI-61026. **H** SEC analysis of wt and stabilized aMPV Pre-F trimer compared with MPV-2c.

interprotomeric salt bridge with R205 and D209E is suboptimal for the ideal salt bridge structure. This suggests that the prefusion structure tolerates certain imperfections around D209, possibly due to structural constraints inherent in the post-fusion conformation. The V231I substitution enhances side chain packing in the hydrophobic pocket shaped by residues 224-252 and the E453P stabilizes the HR2 loop and removes excess negative charge in the cavity on the three-fold axis (Supplementary Fig. 10b, c).

Besides the structural details of the novel Pre-F stabilizing substitutions which were focused on the trimer interface, the double-cleaved trimer with truncated F2 C-terminus without a foldon revealed several unique features related to the increased interprotomeric interactions not seen in previously published structures. (I) Substitutions in HR2 improved the interactions at the trimer interface that resulted in structural elucidation of the full triple helical stem. (II) An extended EM density at N172 suggests an additional glycan residue, potentially forming a glycan 'crown' in the apex with a

potential interaction with the adjacent protomer (Supplementary Fig. 9). (III) Specific interactions involving the novel F2 C-terminus with the neighboring protomer. This includes hydrophobic interactions of L89 and R91 with V38, L243, H332, R329, F334, a hydrogen bond between the L89 main chain and N247, and a salt bridge between R91 and D280 (Fig. 6E, Supplementary Fig. 9).

**Stabilized prefusion hMPV F trimer induces neutralizing and protective immune response.** To evaluate our hMPV Pre-F design as a potential vaccine candidate, the induction of binding and neutralizing antibody titers was compared between the stabilized non-cleaved (MPV-foldon) and the cleaved Pre-F variants (MPV-2c and MPV-2c2) in naïve mice (Fig. 7A, B). Mice immunized with ASO1$_B$ adjuvanted cleaved Pre-F variants generated significantly higher levels of hMPV Pre-F binding antibodies (Fig. 7A), as well as increased levels of neutralizing antibodies (Fig. 7B), when compared to adjuvanted non-cleaved variant. In addition, the presence of hMPV B1 cross-

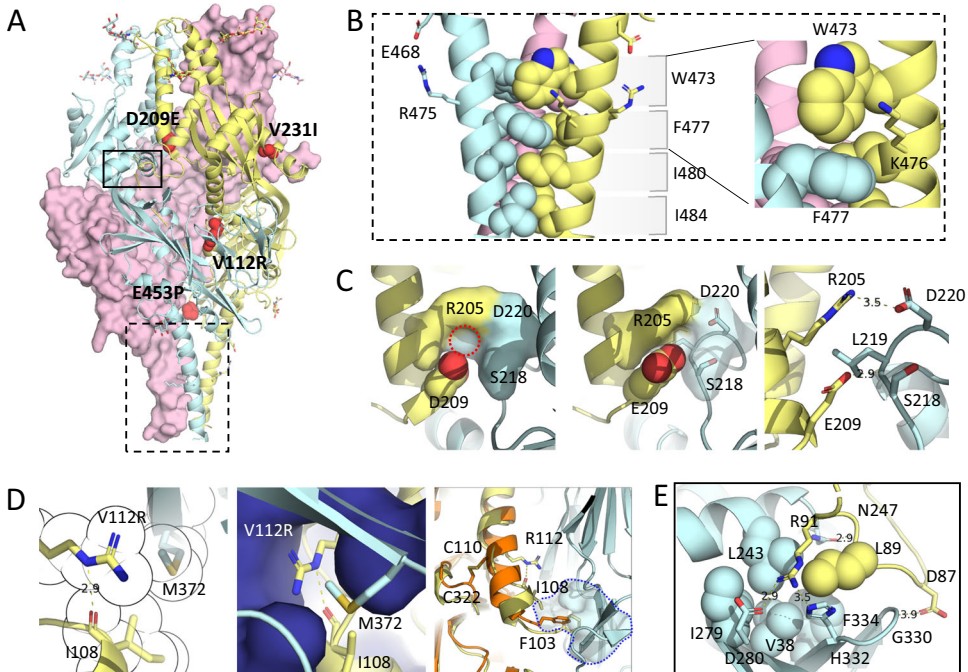

**Fig. 6 | Cryo-EM analysis of stabilized MPV-2c. A** Ribbon structure of stabilized trimer with the three protomers in yellow, blue and pink, respectively. Side chains of stabilizing substitutions plotted in the space-filling representation in red. **B** Side views of stabilized HR2 stem corresponding to the broken rectangle in (A) with the top layer showing the stabilizing W473 residues and the layer beneath showing the stabilizing F477 residues. **C** Zoom in panels of residue D209 (left) and E209 (middle) with interprotomeric interaction at the protomer interface shown as surface in blue and yellow. In case of D209, a small cavity is indicated with the red dotted circle. Side chains are shown for clarity in the right panel. **D** The interaction of residue R112 at protomer interface, highlighting the hydrogen bonding of the R112 guanidinium group with the I108 backbone (left panel). The binding pocket of R112

is depicted with a blue surface (middle panel). The right panel illustrates how R112, located in the fusion peptide, stabilizes the helical kink in the fusion peptide (shown in yellow), influencing its interactions, including the precise positioning of F103 within the binding pocket (indicated by the blue dotted outline). The structure is overlayed with published DS2-Cav-ES2 (in orange) to compare how the engineered C110-C322 disulfide obstructs the helical kink and alters the interaction of the fusion peptide, particularly affecting the key residue F103. **E** The newly created F2 C-terminus of the double-cleaved hMPV Pre-F corresponding to the rectangle in (**A**) shows strong interactions of D87, L89 (yellow spheres) and R91 with a cluster of residues of the adjacent protomer (blue).

neutralizing antibodies could be demonstrated in serum from mice immunized with 15 µg of the hMPV A2-based MPV-2c (Supplementary Fig. 11a). Serum pools of all groups of immunized mice were also tested in a VNA on hAEC cultures in combination with the hMPV CAN97-83-GFP strain. Like the neutralization assay on immortalized cells, also for the hAEC cultures, the serum pools from cleaved Pre-F immunized mice showed the highest neutralizing activity, with viral breakthrough observed at the 1:781 diluted serum (Supplementary Fig. 11b, c). To assess the ability of the stabilized Pre-F protein to boost hMPV pre-existing immune responses, mice were intranasally infected with hMPV A2, and immunized 12 weeks later with 15 µg MPV-2c without adjuvant. At 12 weeks post pre-exposure, hMPV Pre-F binding antibody titers were still measurable, confirming successful pre-exposure, whereas hMPV neutralizing antibody titers were below the detection threshold. Single immunization with unadjuvanted Pre-F protein strongly boosted both Pre-F binding antibodies and neutralizing antibody titers, when measured up to 12 weeks post immunization. (Fig. 7C, D). The ability of MPV-2c to confer protection against hMPV was assessed in an hMPV cotton rat challenge model. Cotton rats were immunized twice with a 3-week interval with MPV-2c adjuvanted with AS01B or with Post-F[40] adjuvanted with AS01B, and intranasally challenged with hMPV A2 3 weeks after the last immunization. Immunization with Pre-F resulted in strong induction of virus neutralizing antibody titers in pre-challenge serum, without vaccine-dose dependent effect. In contrast, neutralizing antibody titers were low in Post-F immunized animals (Fig. 7E). Similar as observed for mice, serum from Pre-F immunized cotton rats also showed cross neutralization against hMPV-B1 (Supplementary Fig. 11a). After hMPV challenge, robust nose and lung viral

loads were observed in mock immunized animals, whereas animals that were intranasally pre-exposed with hMPV at day 0 were completely protected. Animals immunized with MPV-2c showed complete protection in the lower and upper respiratory tract, except for 1 animal that failed to develop antibody titers after immunization. In contrast, animals immunized with Post-F were completely protected in the lungs, whereas only partial protection in the nose was observed (Fig. 7F, G). Lung viral load was additionally assessed using the more sensitive RT-PCR method. Again, strong reduction of viral load was observed in animals immunized with Pre-F, whereas this reduction was less for Post-F immunized animals (Fig. 7H). Correlation analysis showed that there was a clear negative correlation between lung viral load measured by RT-PCR and neutralizing antibody titers in pre-challenge serum (Fig. 7I).

## Discussion

Currently, there is no vaccine or specific antiviral treatment available to prevent hMPV illness, creating a significant unmet need. However, the recent success and regulatory approval of RSV vaccines by the US Food & Drug Administration holds promise that development of a Pre-F protein-based vaccine approach is also possible for hMPV[17,18,41,42]. Here we describe an AI-guided design for a double cleaved, stabilized and closed hMPV Pre-F trimer without a foldon trimerization domain with high expression levels that induces strong protection in cotton rats.

A notable difference between RSV and hMPV F proteins lies in the extended length of the fusion peptide and the F2 C-terminus in hMPV F, implying that the full F2 C-terminal region may not

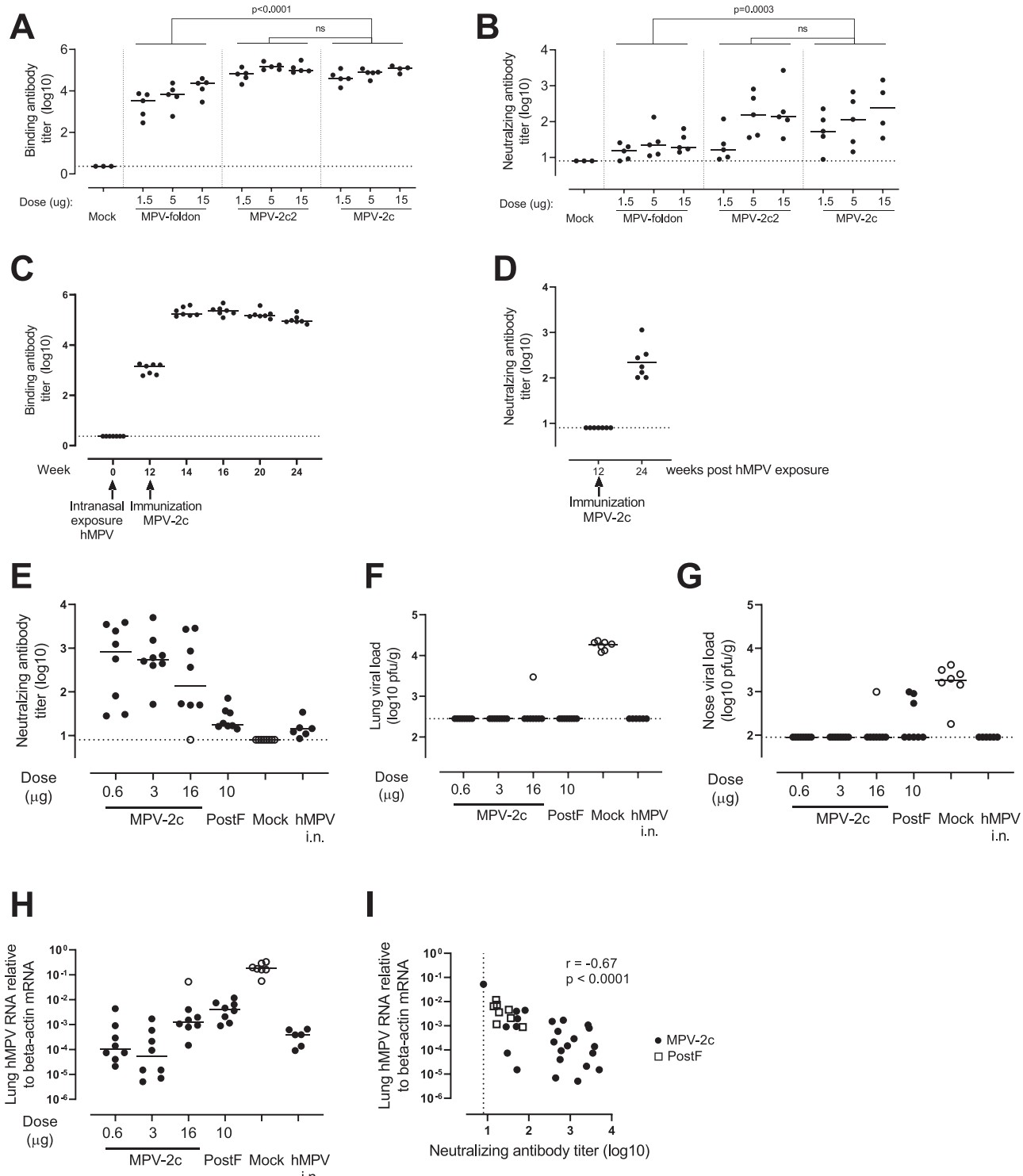

properly fit within the trimer cavity. This hypothesis was supported by our recombinant protein analysis, which showed that truncating the F2 C-terminus of hMPV F by nine amino acids led to the efficient expression of a closed Pre-F trimer. Furthermore, the incorporation of the RSV p27 domain allowed for efficient production and complete double cleavage of hMPV F trimers by ubiquitous furin-like proteases.

The mechanism by which Pre-F trimer achieves closure in the viral context is unclear. The F2 C-terminus has never been observed in the trimer cavity and it obstructs trimerization based on the structures that were solved as monomers[17,31]. Therefore, the F2 C-terminus may

need to be dislocated to the outside of the trimer to allow full closure of the trimer. Alternatively, a second, as of yet unknown, cleavage event in the hMPV F2 C-terminus could allow efficient full closure of the trimer. This would unify the maturation of pneumovirus F proteins, since RSV F also utilizes a double cleavage process during which the p27 peptide gets released allowing trimerization.

Interestingly, several mutations at the F2 C-terminus (E93V, Q94R) are observed in strains that are adapted to cell culture[43,44]. Recently, resistance-conferring mutations were found at A90V, Q94R and E96K after selection with an interface-specific monoclonal antibody, suggesting that these mutations would somehow facilitate closure of the

**Fig. 7 | Immunogenicity and protective efficacy of hMPV Pre-F immunogens in rodent models. A, B** Naive Balb/c mice were immunized at day 0 and day 28 with 1.5, 5 or 15 µg of MPV-foldon, MPV-2c or MPV-2c2, all adjuvanted with AS01b (n = 5 for experimental groups, n = 3 for mock group). Serum isolated at day 42 was analyzed for pre-F binding antibodies by ELISA (**A**), and neutralizing antibodies against hMPV-A2 (**B**). Statistical testing was done using a Tobit model with a Bonferroni correction for multiple comparisons, containing vaccine candidate and dose as explanatory factors (2-sided). **C, D** Balb/c mice (n = 7) were intranasally exposed to hMPV-A2 at week 0 and immunized with 15 µg unadjuvanted MPV-2c at week 12. Serum isolated during the course of the study was analyzed for Pre-F binding antibodies (**C**) and hMPV A2 neutralizing antibodies (**D**) at the timepoints indicated. **E–I** Cotton rats were immunized at day 0 and day 21 with formulation buffer (Mock, n = 9), with 0.6, 3 or 16 µg MPV-2c (n = 8), or with 10 µg hMPV Post-F protein (n = 8), all adjuvanted with AS01b, or received intranasal pre-exposure with hMPV-A2 at day 0 (n = 6; indicated with hMPV i.n.). At day 42, animals were intranasally challenged with hMPV-A2, and viral load was determined 4 days post infection at day 46. hMPV-A2 neutralizing antibody titers against hMPV A2 were determined in pre-challenge serum samples (**E**). Viral load was determined by plaque assay in lung (**F**) and nose (**G**) homogenates, and by RT-PCR in lung homogenates (**H**). Graphical representation of the correlation between lung viral load as measured with RT-PCR, and the neutralizing antibody titers in pre-challenge serum (**I**). Spearman rho (r) correlation coefficient is indicated in the Figure, the testing was two-sided. One animal immunized with 16 µg MPV-2c failed to mount an immune response, and is depicted with an open circle in **E** to **H**. Every dot depicts the value of an individual animal, and the horizontal line indicates the median response of the group. Lower limits of detection are indicated with dotted lines.

trimer and concomitant occlusion of the epitope by allowing more efficient proteolytic cleavage at a potential second cleavage site[45].

While the exact mechanism remains uncertain, our study provided insights that lend support to the double cleavage hypothesis. Unlike previously described structures, the Cryo-EM structures of the Pre-F trimers in this study show a distinct interprotomeric interaction involving the novel F2 C-terminus. This interaction occurs through residues L89 and R91 with a pocket conserved in both avian and human MPV.

Although additional cleavage was observed by mass spectrometry in hMPV produced in hAEC cultures, a dominant second cleavage site could not be identified. This finding encourages further investigation into the specific level, stage or cell type where this postulated second cleavage event might occur. The second cleavage could take place at a later stage during viral infection, or it may not be necessary for all F trimers or all protomers in a trimer.

Independent of the mechanism, open F trimers might induce antibodies with lower neutralizing potential, allowing evasion of an optimal immune response. Although antibodies isolated from healthy human subjects infected with hMPV and a single-domain antibody isolated from an immunized llama can bind to epitopes on the trimeric interface, which are occluded on a closed trimer[29–31,45,46], these antibodies have a very low neutralization potential compared to e.g. the site Ø – specific ADI-61026 (Supplementary Fig. 3). Therefore, in line with our animal study, a closed Pre-F trimer would be a desirable vaccine component, particularly for pre-immune adult populations, to boost potent site Ø antibodies instead of trimer interface antibodies[47]. The most distinctive features of MPV-2c compared with previously described Pre-F proteins is the absence of a heterologous trimerization domain and the increased trimeric stability and low binding to anti-interface antibodies. The hMPV Pre-F structure 5WB0, which is the first reported structure of its kind, was stabilized primarily by a proline substitution in the hinge loop, which yielded limited stability (Fig. 4B), and exhibited immunogenicity similar to Post-F[15]. It also featured a distinctive fusion peptide structure with an N-terminal helical kink influencing the orientation of the trimer stabilizing residue F103, akin to the Pre-F designs in our study. However, later Pre-F protein designs with three additional disulfides for increased stability did not maintain the interaction with the F2 C-terminus and altered the N-terminal fusion peptide's conformation, unlike our fully cleaved and closed MPV-2c2 trimer, which also demonstrated both higher expression and thermostability (Fig. 6)[17,18].

Like other class I fusion proteins, the hMPV F's prefusion conformation is metastable and requires stabilization to improve vaccine efficacy and manufacturability. Since we used the double cleaved backbone allowing for efficient closure of the trimer, we focused on identifying novel stabilizing substitution at the protomer interface. For this purpose, we developed an AI convolutional classifier, called ReCaP, employing it alongside Rosetta's fixed-backbone design[35] and coupled moves methods[36]. Given the insufficient data available to train a classifier for predicting stabilizing mutations, we adopted a different strategy. Our model was trained on a substantial portion of the Protein Data Bank to recognize masked wild-type residues based on their atomic microenvironments. We then repurposed the model to identify unusual microenvironments where wild-type residues were not predicted, and instead, biochemically distinct residues were suggested. We interpreted these microenvironments as potential instabilities and tested the suggested substitutions for their stabilizing effects. The model, though physics-agnostic, aligned with Rosetta in identifying hydrophobic space-filling substitutions but differed in detecting substitutions in the trimer's hydrophilic interior and at protomer interfaces. Notably, the most effective stabilizing substitutions in the Pre-F trimer interface, V112R and D209E, were identified by ReCaP and were not predicted by Rosetta. V112R in the fusion peptide significantly increased Pre-F stability and expression levels, demonstrating the effectiveness of enhancing interactions between the fusion peptide and its binding pocket. The mutation's strong stabilizing effect, despite not being a typical space-filling substitution and lacking nearby counterbalancing charges, can likely be attributed to inhomogeneous solvation effects. Similarly, D209E resulted in substantial stabilization, not readily apparent from structural analysis alone. Both substitutions enhance the geometry of complex polar and charge interactions which are challenging to model or predict through visual inspection, owing to the complexities of the discrete nature of solvation thermodynamics at protein interfaces. Inhomogeneous solvation, well-documented in small molecule ligand binding[48,49], is also critical in protein mutations, particularly where internal cavities and water molecules displacement are involved. ReCaP's predictions suggest its capability to capture such desolvation-related features.

ReCaP differs from a recently described convolutional model[50], since it incorporates Residual Neural Network (ResNet) connections, significantly improving its performance. Recent developments in machine learning have shed light on the efficacy of such ResNets[51] in a variety of fields, notably in medical image processing[52], computer vision[51] and natural language processing[53]. ResNets, characterized by their skip connections, facilitate the flow of gradients during training, thus mitigating the vanishing gradient problem common in traditional deep convolutional neural networks, and allowing for deeper models. This architectural innovation enables AI classifiers based on ResNets to perform finer, more nuanced corrections, in complex residue microenvironments. From the perspective of AI technology, our approach, while relatively straightforward, has proven to be exceptionally powerful for protein engineering especially when minimal modification is preferred, like vaccine components. The strategic implementation of single-point mutations, guided by this model, demonstrated its capacity to significantly alter protein expression and bolster trimer stability and affirms the efficacy of straightforward AI-driven strategies in navigating the complex landscape of protein functionality. Although it is challenging to assess the reliability of the model's performance on the task of detecting protein instabilities due to the absence of suitable data, the distinct advantage of using our AI model is its ability to generate novel mutation ideas, particularly those involving polar or

charged amino acids. This capability is highlighted by the V112R mutation, where, unlike our model, Rosetta predicted the wild-type and a homologous leucine. Our approach offers an alternative perspective to the prevailing trends in protein engineering, where the focus frequently lies on designing entire protein domains through the use of generative models[54–56].

In conclusion, this study showed that a double-cleaved closed trimer allowed screening for the pre-F stabilizing substitutions at the correctly folded trimer interface. The effectiveness of the AI model is underscored by its capacity to rectify destabilizing interactions through the introduction of charged residues into the protein core, a type of substitution typically rather unsuccessful for stabilization. Biochemical assays and the Cryo-EM structural analysis confirmed that the postulated F2 C-terminus truncation agrees with the natural folding of the native closed trimer. The resulting stabilized closed hMPV Pre-F trimer induced high titers of broadly neutralizing antibodies in naive and pre-immune mice and induced strong protection in cotton rats against hMPV challenge. The high expression and stability of MPV-2c makes it a promising vaccine antigen and could help in isolating therapeutic antibodies against the closed trimer interface and apex.

## Methods

### Ethical statement
Mouse studies were conducted at Janssen Vaccines and Prevention B.V. according to the Dutch Animal Experimentation Act, and the Guidelines on the Protection of Animals for scientific purposes by the Council of the European Committee after approval by the Centrale Commissie Dierproeven and the Dier Experimenten Commissie of Janssen Vaccines and Prevention B.V. under ethical protocol AVD2130020209424. Cotton rat studies were conducted at Sigmovir Biosystems, Inc. by permission of the Institutional Animal Care and Use Committee (IACUC) of Sigmovir Biosystems, Inc. under IACUC protocol #15.

### Mass spectrometry
Preparation hMPV samples. An hMPV clinical isolate passaged on llc-mk2 cells was grown on HAECs for 4 days. Cells were cultured on an air–liquid interface and a maximum of 24 µL of culture medium could be harvested from a well in a six-well plate. Supernatant was mixed 1:1 with denaturation buffer. The sample was loaded on gel four times where the proteins were separated on reduced SDS-PAGE using MES running buffer. The bands of interest were excised from the gels and the proteins in each SDS-PAGE band were reduced again and subsequently alkylated and digested using Lys-C in the presence of ProteaseMax for 2 h at 37 °C, after which the supernatant containing peptides was collected. An additional extraction was performed with 100% acetonitrile (ACN) and combined with the first supernatant. Finally, the organic solvent was removed in a vacuum centrifuge for 1 hour. A total of 10 µL of peptides was injected and analyzed using nLC-ESI-Q-Orbitrap MS. An UltiMate 3000 RSLCnano (Thermo Scientific) was equipped with a Waters NanoEase *m/z* symmetry C18 100 Å, 5 µm, 180 µm × 20 mm trap column and a Waters NanoEase m/z HSS C18 T3 Column 100 Å, 1.8 µm, 75 µm × 250 mm column. The latter column was directly coupled to an Orbitrap Q Exactive Plus (Thermo Scientific) by a Thermo Scientific EasySpray nanoflow transfer line 20 µm × 500 mm with 7 µm ID Emitter tip. Each sample was loaded on the trap column at a flow rate of 15.00 µL and, subsequently, eluted from the trap column onto the HSS C18 T3 column using a linear gradient from 5 to 50% mobile phase B in 40 min with a flow rate of 300 nL/min–(mobile phase A - 0.1% Formic Acid in water and mobile phase B - 0.1% Formic Acid, 80% acetonitrile in water). Mass spectra were acquired over m/z 240–1400 at 70,000 resolution and data-dependent acquisition selecting the top 7 most abundant precursor ions for fragmentation at a resolution of 17,500 (at 200–2000 *m/z*). For each sample, two-replicate digestion and two injections per replicate were performed.

For the initial data exploration, the generated MS peptide ion data was matched to a custom database containing the protein sequences of the human proteome, hMPV proteins of interest and known contaminants (http://www.thegpm.org/crap/) using Proteome Discoverer v2.5. For quantification of C-terminal F2 peptides, a combination of MaxQuant 2.4.9.0 and Skyline 23.1.0.268 was used. In MaxQuant peptide search, Carbamidomethyl (C) was set as a fixed modification, while Oxidation (M) and Deamidation (NQ) were set as variable modifications, with maximum three modifications allowed per peptide. Lys-C was set as an enzyme with "Semi-specific free C-terminus" digestion mode to enable detection of truncated C-terminal peptides of hMPV F2 protein. In Skyline, a routine for MS1 quantification based on DDA search was used, whereby MaxQuant peptide search results were used for MS1 filtering. Full-scan settings were set as follows: precursor charge states – 2 and 3, peaks – 4, mass accuracy – 20 ppm. All transitions and integration windows were manually checked. To determine the most abundant peptide variant in each SDS-PAGE band, its relative peptide abundance was calculated. The absolute abundances of all detected C-terminal F2 (truncated) peptides were summed and the percentage of each peptide variant was calculated. This relative abundance was calculated between peptides detected within each SDS-PAGE band separately, therefore relative abundances cannot be compared between SDS-PAGE bands analyzed.

### Expression of hMPV F proteins
DNA fragments encoding C-tagged F proteins were synthesized (Genscript) and cloned in the pcDNA2004 expression vector, a modified pcDNA3 plasmid with an enhanced CMV promotor. Culture supernatants for analytical SEC and Differential Scanning Fluorimetry were generated by transient transfection of Expi293F cells at 200 µL scale in 96-half deep well plates at a viable cell (vc) density of 2.5E + 06 vc/mL using the ExpiFectamine 293 transfection kit (Gibco, Thermo-Fisher Scientific). At day three post-transfection, culture supernatants were harvested, clarified by centrifugation (10 min at $400 \times g$) and filtrated (96-well Filter plates, 0.22 µm PVDF membrane, Corning). Protein batches generated for purification were produced in Expi293F suspension cells (300 mL scale). Expi293F cells were cultured in Expi293F Expression medium [+] GlutaMAX (Gibco, ThermoFisher Scientific) and transiently transfected using ExpiFectamine 293 (Gibco, Thermofisher Scientific) according to manufacturer's instructions and 18 h post transfection enhancers 1 and 2 were added (Gibco, ThermoFisher Scientific). Culture supernatants were harvested at day 5 and were clarified by centrifugation, followed by filtration over a 0.2 µm bottle top filter (Corning).

### Purification of hMPV F proteins
From the harvested culture supernatants, F proteins were purified by a two-step protocol using and ÄKTA Avant 25 system (GE Healthcare). Clarified supernatant was loaded on a C-tag XL 5 mL pre-packed column (ThermoFisher Scientific) equilibrated in 20 mM TRIS, 150 mM NaCl, pH 7.4. Elution of the C-tagged proteins was performed using equilibration buffer containing 2 M MgCl₂. Eluted fractions were diluted 1:1 with equilibration buffer to lower MgCl₂ levels and concentrated using an Amicon Ultra 15 30-kDa cutoff filter (Millipore). To further polish the purified protein Size Exclusion Chromatography (SEC) was performed by running a HiLoad Superdex 200 16/600 column (GE Healthcare) equilibrated in 20 mM Tris, 150 mM NaCl, pH 7.4. Peak fractions were pooled and sterile filtrated using a Millex-GV 0.22 µM filter membrane (Millipore Sigma). hMPV F in the Post-Fusion conformation was based on design previously described[40]. C-tagged Post-F was captured as described before, after which SEC was performed by running a Superose 6 pg, XK 16/70 column (GE Healthcare) equilibrated in 20 mM Tris, 150 mM NaCl, pH 7.4. On the obtained pool a TEV digestion was performed using TEV protease kit (New England Biolabs, #P8112S) according to the manufacturer's instructions. Subsequently a

C-tag step, as described before, was performed on the digestion to remove any non-digested material. The pool containing digested hMPV was loaded onto a 1 mL HisTrap HP column (GE Healthcare) equilibrated in 20 mM Tris, 500 mM NaCl pH 7.4. The column was washed with 20 mM Tris, 500 mM NaCl, 5 mM Imidazole pH 7.4 and all flowthrough of the column was collected, as the TEV contained a His-tag and the target protein not. The obtained product was desalted using a 5 mL HiTrap desalting column (GE Healthcare) equilibrated in 20 mM Tris, 150 mM NaCl, pH 7.4 and the pool containing target product was heat-treated for 10 min at 70 °C to ensure the protein was completely transformed to the Post-F conformation[17]. Finally, the protein was polished by SEC using a Superose 6 increase, 10/300 GL column (GE Healthcare) equilibrated in 20 mM Tris, 150 mM NaCl, pH 7.4. Peak fractions were pooled and sterile filtrated using a Millex-GV 0.22 µM filter membrane (Millipore Sigma).

### Size exclusion chromatography and multi-angle light scattering

Harvested cell supernatants of Expi293F cell cultures were analyzed for the presence of expressed hMPV F and the purity of produced proteins was analyzed by analytical size exclusion chromatography (SEC) as described previously[57]. An Ultra-High-Performance Liquid Chromatography (UHPLC) Vanquish system (ThermoFisher Scientific) in combination with a Unix-C SEC-300 15 cm column (Sepax Technologies) with the corresponding guard column (Sepax Technologies) at 25°C was used. For purified proteins a µDAWN instrument (Wyatt Technology), µ T-rEx differential refractometer (Wyatt Technology), and Nanostar DLS reader (Wyatt Technology) were added in line and everything was equilibrated in 150 mM sodium phosphate, 50 mM sodium chloride, pH 7.0 and run at 0.35 mL/min. Applied samples were maximum 10 µg protein or 20 µL in volume. The UV signal of supernatants of non-transfected cells was subtracted from the UV signal of HA transfected cells. Chromeleon 7.2.8.0 software was used to analyze the data and molecular weight, conformation and hydrodynamic radius of purified hMPV F trimers were calculated by Astra 8.0.0.19 software (Wyatt). In Astra a value of 0.185 was used as dn/dc value for the protein component for the glycan component a value of 0.1410 was used. The RI detector was used as a source for concentration to calculate molecular weights and the UV as concentration source to calculate mass recoveries.

### Heat-SEC on cell supernatant

Per temperature point 45 µL of hMPV-containing supernatant was aliquoted in a 500 µL Eppendorf tube. The supernatant was incubated at 58 °C or 63 °C (Eppendorf ThermoMixer C) for 15 min, whereas the control was kept at 4 °C. Subsequently, the samples were centrifuged at 18,000 × g for 10 min. Readout was performed by applying 20 µL sample to a Unix-C SEC-300 15 cm column (Sepax Technologies) with the corresponding guard column (Sepax Technologies) at 25 °C equilibrated in running buffer (150 mM sodium phosphate, 50 mM NaCl, pH 7.0) at 0.35 mL/min on an ultra-high-performance liquid chromatography system (Vanquish, Thermo Scientific). Analytical SEC data were analyzed using Chromeleon 7.2.8.0.

### SDS-PAGE

Samples were prepared by adding 2 µg of protein to a mixture of 4× LDS buffer and 10x reducing agent (both Invitrogen) to which 1x PBS (Gibco) was added to a final volume of 40 µL. Samples were heated for 10 min at 95 °C without shaking in a Thermomixer (Eppendorf). From a Bolt 4−12% Bis-Tris Plus, 10 wells gel (Invitrogen) the combs and white stickers were removed and rinsed with demi water prior to being placed in a bolt gel running system (Invitrogen). The inner and outer chamber were filled with 1× MES running buffer (20× stock, Novex) and 30 µL of sample (1.5 µg protein) and 10 µL SeeBlue Plus2 Prestained Standard (Invitrogen) were loaded. A powerpack basic (Bio-Rad) was connected and the gel was run at 150 V for 35 min. After running the gel

was stained with InstantBlue Coomassie Protein Stain (Abcam) for 1 h before a picture was taken using an Odyssey CLx (Li-Cor).

### Biolayer interferometry using monoclonal antibodies

Antibodies at a concentration of 5 µg/ml in 1× kinetics buffer (Sartorius, cat. #18-1105) were immobilized to anti-hIgG (AHC) sensors (FortéBio, cat. #18−5060) using 96-well black flat-bottom polypropylene microplates (Corning, cat. #3694) in a 30 min step. Supernatant containing hMPV F protein and mock supernatant were also in 96-well black flat-bottom polypropylene microplates. To perform the experiment an Octet RED384 system (FortéBio) was used where the whole protocol was performed at a shaking speed of 1000 rpm at 30 °C. Sensors were activated in 1× kinetic buffer for 600 s, the antibody immobilized for 1800 s, blocking in mock supernatant for 600 s and binding of the hMPV protein in supernatant for 300 s. In case of purified protein, a protein concentration of 20 µg/ml in 1× kinetics buffer was used. The same protocol was applied, only the blocking step in mock was replaced by a baseline step of 600 s in 1× Kb. Data analysis was performed using FortéBio Data Analysis 12.0 software (FortéBio).

### Differential scanning fluorimetry

The stability of the hMPV F proteins was determined by measuring the melting temperature ($Tm_{50}$) using differential scanning fluorimetry (DSF). Sypro Orange Dye 5000x (Invitrogen) was diluted in 1× PBS (Gibco) to obtain a 50x working solution. For each reaction 20 µg of purified protein was mixed with 1× PBS to a final volume of 90 µL, to which 10 µL 50× Sypro Orange was added. Of the 100 µL volume triplo's of 30 µL were distributed in a MicroAmp Fast Optical 96-well plate (ThermoFisher). 1× PBS without protein was used as a negative control. The plate was covered with a MicroAmp Optical Adhesive Film (ThermoFisher) and was subsequently read in a ViiA7 Real-time PCR machine (Applied Biosystems). The measurements were performed with a starting temperature of 25 °C and a final temperature of 95°C (54 °C increase per hour). Melting temperature was expressed as the temperature in which 50% of the protein was melted ($Tm_{50}$) and was derived from the negative first derivative that was plotted as a function of temperature. The melting temperature could be observed as the lowest point in the obtained curve.

### Freeze-thaw analysis

Purified proteins were subjected to freeze-thaw (FT) cycles in their storage buffer (20 mM Tris, 150 mM NaCl, pH 7.4). Samples were diluted with storage buffer to 0.5 mg/ml in a 250 µL volume and subsequently divided into 5 portions of 50 µL in 500 µL Eppendorf tubes. All vials, except the control, were snap-frozen in liquid nitrogen after which they were thawed at room temperature. As soon as a measuring point was reached (1×, 5× or 10× FT) a vial was picked and no longer included in the freeze-thaw cycles. After finishing the freeze-thaw cycles, the samples were analyzed by SEC-MALS, as described above.

### Rosetta Coupled Moves protocol

Mutation design was conducted using the Coupled Moves protocol from Rosetta, detailed at https://www.rosettacommons.org/docs/latest/coupled-moves. We executed the algorithm in two phases, each defined by different settings of the 'walking_perturber_magnitude' parameter: 3.3 in the first phase and 1.8 in the second. The initial phase consisted of 1000 steps, while the subsequent phase comprised 500 steps. We calculated Coupled Moves scores for each of the 20 possible amino acid substitutions at every position within the protein, including the wild type for normalization. Each score was then adjusted by subtracting the score of the wild-type residue. Mutations resulting in relative scores below -2 Rosetta Units (RUs) were considered for synthesis. However, some mutations were excluded due to their surface exposure based on visual inspection of the protein.

## Amino-acid classifier

**Dataset preparation.** Our dataset comprised 201,000 3D protein structures from the RCSB Protein Data Bank (PDB)[58] collected in September 2022. We excluded RNA and DNA-only assemblies, chains shorter than 40 amino acids, structures solved by NMR spectroscopy, or those with resolution above 3.5 Å. The filtered dataset of 175,000 structures, underwent clustering using MMseqs2[59], grouping chains with at least 50% similarity and 30% sequence overlap. This clustering aimed to avoid separating different domains of the same protein and resulted in about 30,000 clusters with a total of 496,000 chains. The then divided the dataset into training (70%) and test (30%) sets, using a grouped-stratified split based on structure resolution. For cross-validation, we further split the training set into five folds ensuring chains from the same cluster were not split across different sets and maintaining consistent distribution of resolution across all sets. For each structure in the datasets and at each residue position within a structure, referred to as the target amino acid, we extracted a cubic volume of $16^3$ Å$^3$ centered on the target amino acid. The boxes were voxelized and the voxels of the target amino acid were masked. For detail of the box voxelization, see SI.

**Model architecture.** The Residue Classification for Protein Design (ReCaP) model (Fig. 4A) employs a three-dimensional Residual Neural Network (3D ResNet)[51] with 12 layers. It begins with a 3D convolutional (3D Conv) layer, with 4 input channels (C, N, O, and S atoms), using $3 \times 3 \times 3$ kernels and a Rectified Linear Unit (ReLU) activation. The architecture includes three residual groups, where the feature count increases to 96, 192, and 288, optimized through hyperparameter tuning and limited by memory constraints. The final processing involves two fully connected (FC) layers with 256 and 20 channels, utilizing ReLU and Softmax activations respectively, the latter generating probability scores for 20 amino acids. Each residual group contains three sequential blocks, each with two 3D Conv layers ($3 \times 3 \times 3$ kernels) connected by ReLU activations. To facilitate information flow, the output of the second 3D Conv layer is added to the input of the residual block via an identity mapping connection (skip connection). The combined signal is then passed to a ReLU activation. Strided 3D Conv layers in the first block of the second and third residual groups reduce spatial dimensions and computational load. For enhanced training stability and generalization, all convolutional layers include batch-normalization (BN) layers (Fig. 4A).

**Model training and inference.** ReCaP was trained in cross-validation on 8 million masked microenvironments, randomly sampled from the first three of five folds to manage training time. To ensure even exposure to all residue types, balanced minibatches of 1024 samples were used, proportionate to amino acid frequency in the dataset. Per-channel normalization was applied to input microenvironments for zero mean and a unit variance across four channels. Following Torng et al.[60], the model's pretext objective was classifying the target amino acid based on its masked microenvironment. The training was conducted on a single NVIDIA® A10 GPU over 4900 iterations until convergence of training loss. We utilized mixed precision training to reduce time, employing cross-entropy loss and an Adam optimizer with an initial learning rate of $3 \times 10^{-4}$. For inference, we averaged predictions from three models trained on different leave-one-out split, forming an ensemble to improve prediction robustness. This ensemble achieved a 0.69 macro-average multiclass accuracy in the wild-type recovery task on 7 million micro-environments from the test set. Hierarchical clustering of the normalized confusion matrix (Supplementary Fig. 4) indicated the model's ability in capturing natural substitution patterns based on shared amino acid biochemical properties. We trained the ReCaP model on the pretext task known as wild-type recovery, which involves predicting wild-type amino acids from masked microenvironments. We then utilized the model's inherent bias towards predicting stable structures to identify mutations that could enhance the stability of these microenvironments. Specifically, when the model predicted a non-homologous amino acid with a significantly low probability of being wild-type–close to 0–it suggested that such a mutation could be included in our subsequent screening efforts. Mutation selection for a protein, involves extracting 20 probability values per position from the model (Fig. 4A), discarding surface positions with relative solvent accessibility over 30%, and focusing on positions where the wild type was not predicted by the model. We apply a probability threshold of 0.1 to select mutants, followed by visually inspection of mutation candidates refining the selection for in-vitro assays.

**Box voxelization.** The boxed volume encompassed all atoms of the neighboring residues, effectively representing the microenvironment of the target amino acid. The length of the cube edge was selected to capture short-range interactions up to the first layer of neighbor atoms of the target residue. This cubic volume is referred to as the microenvironment box. To optimize the visibility of the side chains, we centered the box on the estimated coordinates of the Cβ atom of the target amino acid. To mitigate bias towards Glycine, we calculated the Cβ coordinates for all amino acids by assuming that the Cβ, Cα, N, and C atoms form a tetrahedron. We considered bond angles ∠NCαCβ, ∠CβCαC, and ∠CCαN to be 109.5°, and all bonds to have a length of 1.5 Å. The estimation of Cβ coordinates involved determining the angle γ between the Cβ-Cα vector and its projection onto the NCαC plane. Subsequently, we shifted the Cα coordinates by the bond length in the direction of the unit vector aligned with the projection of Cβ onto the NCαC plane and rotated by γ around its perpendicular axis. To minimize unnecessary variations in the input data, each microenvironment box was oriented such that the Cβ-Cα bond aligned with the (−1,−1,−1) diagonal of a unit cube representing the box. To remove the remaining degree of freedom, we aligned the perpendicular of the Cβ-Cα bond with the (−1,−1,2) vector. Subsequently, we performed voxelization, which involved converting the continuous positions of atoms within the box into discrete 3D points or voxels. During voxelization, we generated four binary volumes to indicate the presence of carbon, nitrogen, oxygen, and sulfur atoms at specific voxels, based on their respective van der Waals radii. We then set to zero the voxels belonging to the atoms of the target amino acid, therefore generating a voxelated masked microenvironment. The voxelated masked volumes were constructed with a spacing of 0.8 Å and contained $20^3$ voxels. Upon voxelization of all the boxes in our datasets, and the removal of volumes presenting identical content, we obtained a total of 125 million masked microenvironments, each consisting of four channels.

**Grid preparation and data acquisition.** 3.5 μL of purified hMPV pre-fusion ternary complex (MPV-2c, or MPV-2cREKR) at 3 mg/ml was applied to the plasma-cleaned (Gatan Solarus) Quantifoil 1.2/1.3 UltraAuFoil holey gold grid, and subsequently vitrified using a Vitrobot Mark IV (FEI Company). Cryo grids were loaded into a Glacios transmission electron microscope (ThermoFisher Scientific) operating in nanoprobe at 200 keV with a Falcon direct electron detector. Images were recorded with EPU (ThermoFisher Scientific) in counting mode. Images were recorded in EER (Electron-event representation) format corresponding to a total dose of 40.0 electrons per Å$^2$. All details corresponding to individual datasets are summarized in Table S1.

**Electron microscopy data processing.** The movies were subjected to beam-induced motion correction, contrast transfer function (CTF) parameters estimation, automated reference particle picking, particle extraction with a box size of 256 pixels, and two-dimensional (2D) classification in CryoSPARC[61] live during the data acquisition. Particle images with clear prefusion features were merged and subjected to Ab

initio 3D reconstruction with C3 symmetry in CryoSPARC. Multiple rounds of optimized 3D heterogeneous refinement yielded two classes with a clear prefusion density. The particles were refined using Non-uniform (UN) and local refinements within CryoSPARC with C3 symmetry. Local resolution was calculated using ResMap[62]. All resolutions were estimated by applying a soft mask around the protein complex density and based on the gold-standard (two halves of data refined independently) FSC (Fourier shell correlation) = 0.143 criterion. Prior to visualization, all density maps were sharpened by applying different negative temperature factors using automated procedures, along with the half maps, were used for model building. The number of particles in each dataset and other details related to data processing are summarized in Supplementary Figs. 8, 9 and Table S1.

**Model building and refinement.** The initial template of the hMPV prefusion complex was derived from a homology-based model calculated by SWISS-MODEL[63]. The model was docked into the EM density map using Chimera[64] and followed by manual adjustment using COOT[65]. Each model was independently subjected to global refinement and minimization in real space using the module phenix.real_space_refine in PHENIX[66] against separate EM half maps with default parameters. The model was refined into a working half map, and improvement of the model was monitored using the free half map. The geometry parameters of the final models were validated in COOT and using MolProbity[67] and EMRinger[68]. These refinements were performed iteratively until no further improvements were observed. The final refinement statistics were provided in Table S1. Model overfitting was evaluated through its refinement against one cryo-EM half map. FSC curves were calculated between the resulting model and the working half map as well as between the resulting model and the free half and full maps for cross-validation. Figures were produced using PyMOL (The PyMOL Molecular Graphics System) and Chimera.

**In vivo study description: Immunogenicity assessment in naïve and HMPV pre-exposed mice.** Naïve Balb/c mice (8-10 weeks of age) were intramuscularly immunized with PBS (Mock) or the indicated doses of the protein variants, adjuvanted with 1 μg AS01b (GlaxoSmithKline, purchased as adjuvant component of Shingrix vaccine) at day 0 and day 28. In addition, Balb/c mice (6−8 weeks of age) were intranasally pre-exposed with $1 \times 10^3$ or $5 \times 10^3$ HMPV A2 strain 05-011607, $1 \times 10^3$ or $5 \times 10^3$ HMPV A2 strain 12-000005, or with $3 \times 10^4$ or $3 \times 10^5$ pfu HMPV A2 CAN97-83-GFP, at 12 weeks prior to intramuscular immunization with the indicated doses of unadjuvanted MPV-2c protein. Serum samples were collected for assessments of humoral responses.

**In vivo study description: Immunogenicity and protective efficacy assessment in cotton rats.** Female cotton rats (Sigmovir Biosystems, Inc., Rockville, MD, USA; 6−8 weeks of age) were immunized intramuscularly at day 0 and day 21 with PBS (Mock) or the indicated protein doses, adjuvanted with 5 μg AS01b, or received intranasal pre-exposure with $1 \times 10^4$ pfu HMPV-A2 strain TN/94-49 at day 0. At day 42, animals were challenged with $1 \times 10^5$ pfu HMPV-A2 strain TN/94-49. Animals were sacrificed 4 days post-challenge and viral load was determined by plaque assay or RT-PCR. Serum samples were collected prior to challenge for readout of humoral immune responses. Cotton rat experiments were performed by Sigmovir Biosystems, Inc.

**Animal housing.** Mice and cotton rats were housed in individual ventilated Green Line type II (501 cm²) or type III (904 cm²) (TECNI-PLAST S.p.A., Italy) cages under controlled environmental conditions with a temperature range of 20 to 24 °C and a humidity of 55 −/+ 10%. A day and night light cycle (12 h/12 h) was maintained. All animal studies were conducted under class 2 biosafety level.

**ELISA.** IgG antibodies binding to HMPV A2 F protein were measured by ELISA. In brief, 96-well plates were coated with Streptavidin by 2 h incubation at 37 °C. After washing with PBS/Tween, the wells were blocked with 1% casein buffer for 1 h at RT, washed again, followed by addition of biotinylated MPV-2c protein and incubation for 1 h at RT. After washing, serially diluted serum was added and incubated for 1 h at RT. hMPV F specific antibodies were detected by HRP-labeled anti-mouse or anti-cotton rat IgG, incubated 1 h at RT, and after washing the wells were developed with ECL substrate. The luminescence signal was measured with Synergy Neo, and Log10 relative potency was calculated to a standard serum sample.

**Microneutralization assay.** HMPV A2 neutralizing titers were determined by a virus neutralization assay infecting Vero cells with GFP expressing HMPV A2 virus. In brief, serially diluted heat-inactivated serum was mixed with HMPV A2 GFP strain CAN97-83 and incubated for 1 h at RT. Subsequently VERO cells in medium with 2 μg/mL trypsin were added to all wells and plates were incubated overnight at 37 °C, 10% CO2. The next day, wells were washed once with warm PBS and medium + 2 μg/mL trypsin was added to the plates. Plates were incubated for an additional 4 days before removal of medium, PBS wash and measurement of the GFP signal with the Synergy Neo. Neutralizing titers were calculated as Log10 IC50 titers using four-parameter logistic regression curve estimation with a common bottom and top. The lower limit of detection was set at the starting dilution of the serum samples (Log10(8) = 0.903).

**Microneutralization assay HMPV B1.** HMPV B1 neutralizing titers were determined by a virus neutralization assay infecting VERO cells with GFP expressing HMPV B1, and CPE based readout. In brief, VERO cells were preseeded one day prior to infection. On day of infection, serially diluted serum samples were mixed with HMPV B1 GFP strain NL/1/99-GFP3 diluted in medium with 2 μg/mL trypsin + Pen/Strep. After 1 h incubation virus/serum mixture was transferred onto the preseeded VERO cells and plates were incubated overnight at 37 °C, 10% CO₂. The next day, wells were washed once with PBS and fresh medium with 2 μg/mL trypsin + Pen/Strep was added to the plates. Plates were incubated for an additional 5 days, after which each well was scored positive or negative for virus infection using the GFP signal. Neutralizing titers were calculated as Log10 IC50 using Spearman Karber.

**Plaque titration assay for hMPV titers in cotton rat challenge model.** Lung and nasal tissues were obtained at 4 days after challenge. Left lung lobes from each cotton rat were weighed prior to homogenization; nasal tissue samples were estimated to be 0.3 g. Plaque titration was performed in duplicates for the homogenized tissue samples on LLC-MK2 cells. After 7 days of incubation, plaques were immunostained and counted to determine the virus concentration in pfu/g of tissue.

**RT-PCR for hMPV lung titers in cotton rat challenge model.** Total RNA was extracted from homogenized lingular lung lobes, and cDNA was generated using a mixture of oligo-dT and random primers (Qiagen QuantiTect Reverse Transcriptase kit). Real-time PCR was performed for the L gene from HMPV-A2 and cotton rat beta-actin using Bio-Rad iQTM SYBR Green Supermix. Amplifications were performed on a Bio-Rad iCycler and baseline cycles and cycle threshold (Ct) were calculated by the iQ5 software. The Ct values of a standard curve were plotted against log10 cDNA dilution factor. These curves were used to convert the Ct values obtained for different samples into relative units. These relative units were then normalized to the level of beta-actin mRNA ("housekeeping gene").

**hAEC culture neutralization assay.** Fully differentiated human airway epithelial cells (hAEC) of nasal origin (MucilAir) and grown on an air-

liquid interface were purchased from Epithelix Sarl (Geneva, Switzerland). The inserts consisted of cells from 14 anonymized donors. The neutralizing capacity of monoclonal antibodies and sere were tested in hAEC's infected by HMPV-GFP. Ready-to-use MucilAir inserts were maintained at an air-liquid interface according to the manufacturer's instructions for 4 days prior to the start of the experiment. Each hAEC insert had undergone prior testing by the manufacturer to ensure ciliary beating, polarization of the epithelial layer and mucus production, as corresponding to healthy respiratory epithelium. At the start of the experiment, inserts were washed once with 200 µL PBS to remove mucus and cell debris. Cells were infected using 10.000 TCID50 units of HMPV-GFP diluted to a final volume of 25 µL 1× PBS, was gently mixed 1:1 with 25 µL antibody or serum dilution, before being added to the apical side of the inserts. After a 1 h incubation at 37 °C, the antibody or serum/virus mixture was aspirated. Negative controls were infected in the presence of a 50-fold dilution of serum from mock-vaccinated mice. Inserts were incubated for 96 h at 37 °C post infection before the GFP fluorescent signal was visualized using a BioTek Cytation 1 automated microscope (Agilent Technologies, CA, USA) using a 2.5× plan achromat objective (Meiji Techno, CA, USA). The individual images were stitched using the Gen5i Plus v3.08.01 software package.

## Reporting summary

Further information on research design is available in the Nature Portfolio Reporting Summary linked to this article.

## Data availability

The coordinates and EM maps generated in this study have been deposited in the Protein Data Bank (PDB) and the Electron Microscopy Data Bank under accession codes: 8VT2, EMD-43516 (MPV-2c), and PDB 8VT3, EMD-43517 (MPV-2cREKR), respectively. MS data are available via ProteomeXchange with identifier PXD049443. Source data are provided with this paper.

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

## Acknowledgements

We thank Niels van den Broek, Sven Blokland, Pascale Bouchier, Cong Liu, Lian Wang, Jan Serroyen and Renske Bolder for technical assistance and Lucy Rutten, Adrian Apetri, Cong Liu, Patrick Forré, Sujata Sharma and Freek Cox for fruitful discussions.

## Author contributions

M.J.G.B., T.R., L.v.d.F., R.Z., and J.P.M.L. designed the study M.J.G.B, T.R., D.v.O., L.L., R.V., M.M.v.H. and M.d.M planned and/or performed biochemical assays and purifications. M.T., L.v.d.F. planned and/or performed preclinical experiments. J.D., A.A.Z., and J.J. developed AI models. X.Y. performed the Cryo-EM characterization. S.T. performed

intact mass LC-MS analysis. M.J.G.B., L.v.d.F., X.Y., J.J., and J.P.M.L wrote the paper with input from all other authors.

## Competing interests

M.J.G.B., T.R., J.J., and J.P.M.L. are co-inventors on related vaccine patents WO2023110618 and WO2023217988. M.J.G.B., T.R., M.T., J.D., A.A.Z., X.Y., D.vO., L.L., R.V., M.M.v.H., M.d.M., S.T., L.v.d.F., R.Z., J.J. and J.P.M.L. are past or present employees of Johnson and Johnson and may hold stocks. The remaining authors declare no other competing financial or non-financial interests.
