## [Peer Review File · Nature Communications]

Efficacious human metapneumovirus vaccine based on AI-guided engineering of a closed prefusion trimerREVIEWER COMMENTS

Reviewer #1 (Remarks to the Author):

In “Efficacious human metapneumovirus vaccine based on AI-guided engineering of a closed prefusion trimer”, Bakker, Langedijk, and colleagues analyze mMPV F processing, showing complete processing from fully differentiated human airway epithelial cells, but incomplete processing of vaccine immunogens. They stabilized the C-terminal region with cavity-filling mutations, with highest trimer yields resulted when F was cleaved, F2-truncated, and HR2 stem region stabilized to form a variant named “hMPV-F-trunc”. Bakker et al. further stabilize this construct by either standard Rosetta or by a novel AI-based approaches, with the latter yielding a particular well expressed trimer that showed good antigenic reactivity, and the authors determined the structures of two variants that differed in the furin cleavage site by single-particle cryo-EM. The authors delineate several unique features related to the increased interprotomeric interactions not seen in previously published structures.

The authors test the immunogenicity of two of their hMPV variants in naïve mice, finding significantly higher titers than induced by stabilized non-cleaved (MPV-foldon) version. The authors also demonstrate high titers by MPV-2c F variant in a pre-exposed model, and further for protection with hMPV cotton rat challenge, that were higher than for postfusion.

Overall, a wonderful paper, investigating not only HMVP F cleavage, but also the impact of stabilization on expression and immunogenicity to yield an improved prefusion-closed HMPV F trimer immunogens.

A few suggestions to improve:

1. In structures showing stabilization – the authors should highlight residues that have been altered from wild-type. Thus for example in Supplemental Figure 11, it would be helpful to see which of the residues have been altered to stabilize the prefusion closed state; if none, then the authors should state this.
2. Second, in both naïve and pre-exposure experiments, it would be helpful to cite the immunogenicity from a control, such as postfusion F for reference to allow comparison with other published papers. (To avoid having to repeat, in naïve, there is “MPV-foldon” – so at minimum, noting in figure legend how MPV-foldon performs relative to postfusion would at least provide readers with a ball-park comparison.)
3. The authors should compare more comprehensively the structure of the fully cleaved, stabilized. MPV-2c F variant with other prefusion stabilized hMPV trimers, such as the initial prefusion stabilized trimer that elicited neutralizing titers similar to the postfusion form (Battles 2017) and the single-chain version recently described by Ou et al (2023), whose structure with MPE8 appears to be in the closed conformation and immunization of which also yields much higher MPV-neutralizing titers than the postfusion form. The structural similarities/differences between these other trimers is difficult to infer from what is presented: it would substantially improve the manuscript to add a supplemental figure with a detailed structural comparison: delineating alterations in construct sequences and resultant residue-level structures, and linking these to impact on immunogenicity.

Reviewer #2 (Remarks to the Author):

The manuscript describes multiple layers of optimization of the F protein of hMPV and its stabilization. Analyzed candidates are narrowed down, neutralization and challenge studies are presented, and a cryoEM-based structure of the most leading candidate is determined. A substantial improvement of stability as measured through melting temperatures, SEC and antibody binding verifying robust trimerization is presented. The two strategies are based on introducing a second cleavage site with the reasoning that more space “within” the trimer cavity would be freed. New trypsin sites were integrated, however, the solution that worked was the transfer of the p27 domain. Sequence optimization is then undertaken focusing on the core and HR2 region and introduction of polar residues identified from comparison to the RSV F protein.

While the data indicates that the authors indeed achieved to stabilize the F protein of hMPV and enabled more solid trimerization of the protein as previously seen, detailed information on how the authors got there in context of substitutions is much needed. Two methods are mentioned that one uses the more biophysics-based approach to designing mutation and Rosetta and a second method that uses a “classifier” AI system which is not much described in detail. No details on the computations with Rosetta are given; many protocols have been previously published, and arguably, running a single design protocol would immediately reveal stabilizing mutations. Similarly, no insight for the classifier is given. If this is indeed the first time this software is used, a detailed description of how it works and what has been done to verify that its output is reliable and we are not looking at a one-off? Since it is a classifier, what were the labels? How does it suggest substitutions, what is the output? how many were tried? The SI has some more details but that should be also integrated in some form in main text and figures.

While that may be harder to realize at this point, why aren't there hMPV exposure controls for the cotton rat experiments?

A minor thought, what is the column used, should be more apparent in figure? It would be easier for readers to work with elution volume as flow rate isn't even in figure legend.

Figure 3 plots: labels are hard to follow.

Figure 4 zoom in isn't helping, needs to be bigger or bigger in figure.

Reviewer #3 (Remarks to the Author):

Bakkers et al. report on a structure-guided design of a stabzed prefusion trimer of the human MPV F protein. They used an AI facilitated approach to achieve a novel design that proved to high highly immunogenic and protective in an animal model. The paper is well written and the conclusions are

justified by the findings.

Comments

- 1) Supplemental figure 1 – the diversity of C-terminal sequences of F2 suggests carboxypeptidase activity may be degrading the peptides. Was there an effort to inhibit carboxypeptidase activity to help identify the exact cleavage site.
- 2) Figure 7H – Why was the high dose (16ug) of MPV-2c less protective than 3ug or 0.6ug.
- 3) In Figures 7H and 7I the labeling and units for viral load are not clear.
- 4) Figure 7I – if the post-F data were not included it looks like the correlation between PCR viral load and neutralizing activity would not be very strong.

Minor comments

- 1) Line 62 – in some RSV F structures a peptide fragment on the outside can be seen attached to one of the protomers. This raises the question whether all 3 protomers have to be cleaved to achieve a fully closed trimer.
- 2) Line 91 – The fact that hAECs produce fully cleaved trimers whereas cells transduced with TMPRSS2 do not achieve full cleavage suggest more than TMPRSS2 is needed for the second cleavage. Do the investigators know more about additional determinants of natural cleavage?
- 3) Line 317 – Was immunogenicity checked with and without the N172 glycan?
- 4) Line 424 – “also” instead of “alco”
- 5) Line 806 – Was the commercial adjuvant AS01b used or was an analogue used? I don't see mention of a GSK collaboration.

original reviewer's comments in Blue

Reviewer #1 Comments

In “Efficacious human metapneumovirus vaccine based on AI-guided engineering of a closed prefusion trimer”, Bakker, Langedijk, and colleagues analyze mMPV F processing, showing complete processing from fully differentiated human airway epithelial cells, but incomplete processing of vaccine immunogens. They stabilized the C-terminal region with cavity-filling mutations, with highest trimer yields resulted when F was cleaved, F2-truncated, and HR2 stem region stabilized to form a variant named “hMPV-F-trunc”. Bakker et al. further stabilize this construct by either standard Rosetta or by a novel AI-based approaches, with the latter yielding a particular well expressed trimer that showed good antigenic reactivity, and the authors determined the structures of two variants that differed in the furin cleavage site by single-particle cryo-EM. The authors delineate several unique features related to the increased interprotomeric interactions not seen in previously published structures.

The authors test the immunogenicity of two of their hMPV variants in naïve mice, finding significantly higher titers than induced by stabilized non-cleaved (MPV-foldon) version. The authors also demonstrate high titers by MPV-2c F variant in a pre-exposed model, and further for protection with hMPV cotton rat challenge, that were higher than for postfusion.

Overall, a wonderful paper, investigating not only HMVP F cleavage, but also the impact of stabilization on expression and immunogenicity to yield an improved prefusion-closed HMPV F trimer immunogens.

We would like to thank the reviewer for these kind words

A few suggestions to improve:

1. In structures showing stabilization – the authors should highlight residues that have been altered from wild-type. Thus for example in Supplemental Figure 11, it would be helpful to see which of the residues have been altered to stabilize the prefusion closed state; if none, then the authors should state this.

In Figure 6 all substitutions are indicated as red spheres except for the substitutions in HR2 since it compromised the visibility. Therefore we included an additional panel (B) which shows all substitutions in HR2. In panel B the color of the chain is chosen to indicate the interactions along the three-fold axis. In order to make the panels in Supplemental Figure 11 (current Figure S10) more clear we have re-colored the indicated substitutions and labels in magenta (Supplementary Figure S10b,c). In figure B, residue V231 was mutated to Isoleucine. In Figure C, residue E453 (left panel) was mutated to Proline (right panel). We also show the wild-type structure to indicate the proximity of the negatively charged residues, an inter-trimeric repulsion that was remediated by mutation. We included titles for all panels to indicate the structure that is visualized.

Second, in both naïve and pre-exposure experiments, it would be helpful to cite the immunogenicity from a control, such as postfusion F for reference to allow comparison with other published papers. (To avoid having to repeat, in naïve, there is “MPV-foldon” – so at minimum, noting in figure legend

how MPV-foldon performs relative to postfusion would at least provide readers with a ball-park comparison.)

The reviewer raises an important point. In this study we did not emphasize comparisons with postF VNA induction since the superiority of PreF over PostF was already described by Hsieh et. al., (2022) and Ou et. al. (2023). Therefore, we did not take along a postF control antigen in our preclinical mouse studies for a side-by-side comparison to the various preF designs. Since this is the first published cotton rat challenge model, we did perform an in-study comparison between adjuvanted PreF and PostF in the naïve cotton rat study that is presented in Figure 7E to 7I.

We did attempt to compare the VNA titer ratio of PreF/PostF calculated for the three different studies that have shown improved VNA induction with PreF vs Post F. These are as follows:

Bakkers et. al.: 30-fold (cotton rat)

Ou et. al., : 20-fold (mouse)

Hsieh et. al., 6-fold (mouse)

However, since the different studies were performed with different dosages, in different species, with different neutralization assays and in different labs, we think that such a comparison should be interpreted with great caution. We therefore preferred to describe that our PreF induced high and cross-neutralizing antibody responses, and the fact that this study is the first that shows near complete protection against hMPV challenge in cotton rats.

Importantly, although the induction of cross-reactive neutralization and protection looks very good for our PreF design, the most important novelty from a vaccine perspective is the high quality, stability and concomitant very high expression levels of our design which is of crucial importance for vaccine development.

3. The authors should compare more comprehensively the structure of the fully cleaved, stabilized. MPV-2c F variant with other prefusion stabilized hMPV trimers, such as the initial prefusion stabilized trimer that elicited neutralizing titers similar to the postfusion form (Battles 2017) and the single-chain version recently described by Ou et al (2023), whose structure with MPE8 appears to be in the closed conformation and immunization of which also yields much higher MPV-neutralizing titers than the postfusion form. The structural similarities/differences between these other trimers is difficult to infer from what is presented: it would substantially improve the manuscript to add a supplemental figure with a detailed structural comparison: delineating alterations in construct sequences and resultant residue-level structures, and linking these to impact on immunogenicity.

We included a structural comparison of MPV-2c with the prefusion structures of Battles (2017), Hsieh (2022) and Ou (2023) in Supplementary Figure 10a. The most striking difference is the elucidation of the full HR2 stem region for MPV2c compared with the other foldon containing F proteins and the observation that the Hsieh (2022) structure is monomeric (Supplementary Figure S10a). Another important difference between MPV-2c and the other designs are the quality aspects that likely impact the immunogenicity, like stability and dynamics (binding to anti-interface antibodies) but these aspects are structurally more subtle. Several of these subtle structural difference have been described in the Result section and Discussion section (deletion of foldon and improvement of HR2 interface interaction and specific interprotomeric interactions of the novel F2 C-terminus with the neighboring protomer)

We have included an additional section in the Discussion in an effort to explain the impact of the current structure on immunogenicity in which we included the difference concerning the interprotomeric interactions of the fusion peptide (line 419):

The most distinctive features of MPV-2c compared with previously described Pre-F proteins is the absence of a heterologous trimerization domain and the increased trimeric stability and low binding to anti-interface antibodies. The hMPV Pre-F structure 5WB0, which is the first reported structure of its kind, was stabilized primarily by a proline substitution in the hinge loop, which yielded limited stability (Fig. 4b), and exhibited immunogenicity similar to Post-F (Battles et al., 2017). It also featured a distinctive fusion peptide structure with an N-terminal helical kink influencing the orientation of the trimer stabilizing residue F103, akin to the Pre-F designs in our study. However, later Pre-F protein designs with three additional disulfides for increased stability did not maintain the interaction with the F2 C-terminus and altered the N-terminal fusion peptide's conformation, unlike our fully cleaved and closed MPV-2c2 trimer, which also demonstrated both higher expression and thermostability (Fig. 6d) (Hsieh et al., 2022, Ou et al., 2023).

Reviewer #2 Comments

The manuscript describes multiple layers of optimization of the F protein of hMPV and its stabilization. Analyzed candidates are narrowed down, neutralization and challenge studies are presented, and a cryoEM-based structure of the most leading candidate is determined. A substantial improvement of stability as measured through melting temperatures, SEC and antibody binding verifying robust trimerization is presented. The two strategies are based on introducing a second cleavage site with the reasoning that more space “within” the trimer cavity would be freed. New trypsin sites were integrated, however, the solution that worked was the transfer of the p27 domain. Sequence optimization is then undertaken focusing on the core and HR2 region and introduction of polar residues identified from comparison to the RSV F protein.

While the data indicates that the authors indeed achieved to stabilize the F protein of hMPV and enabled more solid trimerization of the protein as previously seen, detailed information on how the authors got there in context of substitutions is much needed. Two methods are mentioned that one uses the more biophysics-based approach to designing mutation and Rosetta and a second method that uses a “classifier” AI system which is not much described in detail.

We improved the details of the AI model, including dataset preparation, training and inference are described in the methods section. We have added in the main text: “For details of model training and inference see Materials and Methods section.” In the materials and methods section – model training and inference, we expanded on how the wild-type residue prediction model is used for the purpose of stabilizing the protein: “ We trained the ReCaP model on the pretext task known as wild-type recovery, which involves predicting wild-type amino acids from masked microenvironments. We then utilized the model's inherent bias towards predicting stable structures to identify substitutions that could enhance the stability of these microenvironments. Specifically, when the model predicted a non-homologous amino acid with a significantly low probability of being wild-type—close to 0—it suggested that such a substitution could be included in our subsequent screening efforts.”

No details on the computations with Rosetta are given; many protocols have been previously published, and arguably, running a single design protocol would immediately reveal stabilizing mutations.

To design mutations with Rosetta, we employed the Coupled Moves algorithm (referenced in 35, 36). In the past, our protocols utilized a fixed backbone design, which did not account for backbone flexibility. The Coupled Moves approach represents a more sophisticated advancement of this method by incorporating backbone dynamics, offering a nuanced improvement. However, it is important to acknowledge that no computational approach can predict stabilizing mutations with complete accuracy. The variability in hit rates, typically ranging from 5 to 15%, can be attributed to several factors. The Rosetta energy function, for instance, is an approximation and does not account for inhomogeneous solvation effects. Additionally, the variability is also due to the fact that not all sequences express effectively. Given these constraints and in response to the reviewer's request, we have added a new section to the Materials and Methods titled "Rosetta Coupled Moves Protocol." This section provides a detailed explanation of how we applied the protocol, including specific parameter settings used in our study. Importantly, our model has identified successful mutations like V112R and D209E, which Rosetta did not predict. This success likely stems from the model's ability to account for complex interactions within the protein's microenvironment, particularly inhomogeneous solvation that Rosetta's energy function does not consider. Rosetta excels in predicting shape complementarity mutations where desolvation energies are more straightforward to estimate. Thus, the distinct advantage of using our AI model lies in its generation of novel mutation ideas, especially involving polar or charged amino acids, which are typically not well-predicted by physics-based models like Rosetta. This underscores the AI model's utility in proposing innovative and potentially valuable mutations for further exploration.

Similarly, no insight for the classifier is given. If this is indeed the first time this software is used, a detailed description of how it works and what has been done to verify that its output is reliable and we are not looking at a one-off?

We appreciate the reviewer highlighting the crucial issue of reproducibility. Our model demonstrates reliable outputs for wild-type prediction accuracy. However, applying it to other tasks such as predicting stabilizing mutations (outliers) is challenging to quantify due to a lack of comparative data. Relying on literature for stabilizing mutations is not feasible as the majority are predicted using Rosetta, introducing inherent biases. A comprehensive assessment comparing the AI model's predictions against Rosetta across multiple proteins would require an extensive, unbiased screening. Such an undertaking is beyond the current study's scope. Nonetheless, we have applied the model to several unpublished internal projects, where it has successfully identified diverse mutation types not suggested by Rosetta. To make this aspect of our approach a bit more clear, we added in the Discussion: "Given the insufficient data available to train a classifier for predicting stabilizing mutations, we adopted a different strategy. Our model was trained on a substantial portion of the Protein Data Bank to recognize masked wild-type residues based on their atomic microenvironments. We then repurposed the model to identify unusual microenvironments where wild-type residues were not predicted, and instead, biochemically distinct residues were suggested. We interpreted these microenvironments as potential instabilities and tested the suggested mutations for their stabilizing effects."

At another place we have also added: "Although it is challenging to assess the reliability of the model's performance on the task of detecting protein instabilities due to the absence of suitable data,

the distinct advantage of using our AI model is its ability to generate novel mutation ideas, particularly those involving polar or charged amino acids. This capability is highlighted by the V112R mutation, where, unlike our model, Rosetta predicted the wild-type and a homologous leucine."

Since it is a classifier, what were the labels?

We thank the reviewer for pointing out the lack of clarity in explaining the architecture of our classifier. We have modified the text in the main text: "The classifier, built as a residual convolutional neural network (Fig. 4a) and designed to predict masked amino-acids based on their voxelized atomic micro-environments, was trained on a large dataset of protein structures from the Protein Data Bank (PDB), effectively enabling it to recognize typical, stable and relatively well expressing protein structure. "

How does it suggest substitutions, what is the output?

In the answer to reviewer's first question on details of the AI model, we have expanded on the inference, in the "Model training and inference" section in Materials and methods. We underscore that the model was trained on a different task that it was used for. The model was trained to predict a missing amino-acid in a box of atoms. The output of the model is a vector of 20 probabilities, each saying how likely it is that the amino-acid under consideration is the one that is missing in the box. Majority of amino-acids are predicted correctly, with wild-type having the largest probability in the probability vector. In a large percentage, a homologous amino-acid is predicted, which is to be expected. IN a smaller percent of cases, a biochemically very different amino-acid is predicted, and wild-type is not. We believe this is an indication of an unstable micro-environment and the unusual mutation is a stabilizing mutation.

To make this more clear we have included in the methods section: "We trained the ReCaP model on the pretext task known as wild-type recovery, which involves predicting wild-type amino acids from masked microenvironments. We then utilized the model's inherent bias towards predicting stable structures to identify mutations that could enhance the stability of these microenvironments. Specifically, when the model predicted a non-homologous amino acid with a significantly low probability of being wild-type—close to 0—it suggested that such a mutation could be included in our subsequent screening efforts." We have also added in the main text: "For the purpose of identifying stabilizing mutations, we used all 20 amino-acid probabilities from the model's output (See Figure 4)."

how many were tried?

We have evaluated several models, primarily to determine the optimal resolution for voxelization. In our methodology, the size of the input data increases inversely with the cube of the resolution. Therefore, we aimed to establish the lowest resolution threshold at which further increases would not enhance the accuracy of wild-type predictions. Additionally, we experimented with incorporating atomic charges along with atom types, but this modification did not yield any improvement in performance. We also explored more complex architectures by adding additional convolutional layers; however, these adjustments failed to enhance performance as well.

In order to address this, we added in the main text: “We evaluated various models to optimize voxelization resolution, discovering that reductions below the selected threshold of 0.8Å did not enhance wild-type prediction accuracy. Additionally, attempts to improve performance by incorporating atomic charges and adding more convolutional layers were unsuccessful.”

In case the reviewer is asking about how many mutations were tested, they are all listed in Figure 4A. To make this clearer we have added in the legend of Figure 4: “The libraries designed with Rosetta and the AI model are listed in black and red frames respectively.”

The SI has some more details but that should be also integrated in some form in main text and figures.

We have now incorporated supplementary figure 4 into main figure 4, and updated the figure legend with more detail about the neural network architecture: “The neural network architecture includes a feature extraction component, constructed from convolutional layers equipped with skip connections and the ReLU activation function. Batch normalization (BN) is implemented in the second residual block to enhance stability and performance. The network ends with a fully connected classification layer, followed by a softmax function that normalizes the output into probabilities across the 20 classes.” Additionally, we made the reference to the Supplementary method describing voxelization clearer.

While that may be harder to realize at this point, why aren't there hMPV exposure controls for the cotton rat experiments?

In the cotton rat study presented in the manuscript, a control group was taken along that was intranasally exposed with a low dose of hMPV at day 0, prior to challenge at day 42. Results of that control group are included in the revised version of the manuscript (Figures 7E to 7H), and the text was adjusted accordingly.

A minor thought, what is the column used, should be more apparent in figure? It would be easier for readers to work with elution volume as flow rate isn't even in figure legend.

We fully agree with the reviewer and have changed the axis of the SEC graphs to reflect the elution volume in all main and supplemental figures. Details regarding the exact system and column used for analytical SEC are given in the methods section, and, as we believe adding them to the figure legends might make them unnecessary lengthy, we prefer to keep them as is.

Figure 3 plots: labels are hard to follow.

The plots of Figure 3 were made for easy visualization of the raw SEC files and octet measurements with two antibodies. For reference all the raw data is shown in Supplementary figure 2. We agree with the reviewer that the identity of the labels in the plot is rather schematic and may therefore be difficult to follow. Therefore, we included a representation of the F protein in the top right corner with the three design elements highlighted.

Figure 4 zoom in isn't helping, needs to be bigger or bigger in figure.

We agree that the zoom-in in the present form was not adding much, and, with the raw data present in the supplementals anyway, was actually rather superfluous. In the updated manuscript we decided to remove the zoom-in altogether.

Reviewer #3 (Remarks to the Author):

Bakkers et al. report on a structure-guided design of a stabzed prefusion trimer of the human MPV F protein. They used an AI facilitated approach to achieve a novel design that proved to high highly immunogenic and protective in an animal model. The paper is well written and the conclusions are justified by the findings.

Comments

1) Supplemental figure 1 – the diversity of C-terminal sequences of F2 suggests carboxypeptidase activity may be degrading the peptides. Was there an effort to inhibit carboxypeptidase activity to help identify the exact cleavage site.

The reviewer is right, the C-terminal sequence diversity could indicate carboxypeptidase activity but we have not pursued additional studies to identify the exact cleavage site. We were excited to see that the F protein was completely cleaved in the HAEC cultures in contrast to immortalized cells. However, we were expecting to find a more dominant second cleavage site. Inhibiting carboxypeptidases to reduce the background proteolysis might be helpful in that respect but without a dominant second cleavage site to begin with, this may not be successful. As discussed in the Discussion section, a possible first step could be to investigate into the specific level, stage or cell type where this postulated second cleavage event might occur, in order to see if an enriched dominant second cleavage site could be identified before investigating the possible role of carboxypeptidases.

2) Figure 7H – Why was the high dose (16ug) of MPV-2c less protective than 3ug or 0.6ug.

This was indeed an intriguing observation made in this study: the highest vaccine dose of 16ug appeared less immunogenic and less protective when compared to the lower doses used. However, the lack of a true dose-response is not completely uncommon in animal studies, and more often a so-called hook-effect can be observed. It has been hypothesized that this is due to a (biological) plateau in the induced immune response. We felt that extensive discussion of this observation was outside the scope of current manuscript.

3) In Figures 7H and 7I the labeling and units for viral load are not clear.

This has been adapted in the revised manuscript.

4) Figure 7I – if the post-F data were not included it looks like the correlation between PCR viral load and neutralizing activity would not be very strong.

We agree with the reviewer that the correlation between PCR viral load and neutralizing antibody activity is less strong when the postF data are excluded from the analysis, although still then, a significant correlation is found. This is mainly due to a limited number of data points of PreF immunized animals with low neutralizing antibody titers. We feel that including the postF data in the correlation analysis is valid and substantiates the analysis. However, extensive speculation on a causative relationship between neutralizing antibodies and protection is avoided in the manuscript.

Minor comments

- 1) Line 62 – in some RSV F structures a peptide fragment on the outside can be seen attached to one of the protomers. This raises the question whether all 3 protomers have to be cleaved to achieve a fully closed trimer.

This is an interesting idea. We did postulate that double cleavage may not be needed for all F trimers, but indeed, it could even be the case that it is not needed for all protomers in the F trimer. Perhaps double cleavage of one protomer would allow two single-cleaved F2 C-terminal peptides in the central cavity (or two double-cleaved would allow one F2 C-terminal peptide inside). We have included this possibility in the current version of the Discussion in line 411.

- 2) Line 91 – The fact that hAECs produce fully cleaved trimers whereas cells transduced with TMPRSS2 do not achieve full cleavage suggest more than TMPRSS2 is needed for the second cleavage. Do the investigators know more about additional determinants of natural cleavage?

This is indeed an interesting observation. We are unaware of additional determinants for natural cleavage. The protease family is large and it is more than likely that hAEC's express additional (TMPRSS2-like) protease. It would be of interest if future studies could look into this using, for example, small molecule or peptide inhibitors of specific proteases.

An alternative explanation might be that co-transfection was not fully successful and some cells expressed F but not TMPRSS2.

- 3) Line 317 – Was immunogenicity checked with and without the N172 glycan?

We have not studied immunogenicity of glycan variants or F trimers without glycan occupancy at position 172. We actually do not want to suggest that we have observed differences in glycan occupancy in our recombinant proteins. The observation described in line 317 does not refer to an additional glycan chain but the structural determination of an additional glycan residue which results in a more detailed resolution of the glycan at position 172. This may be the result of the higher stability and closure of the trimer.

- 4) Line 424 – “also” instead of “alco”

Corrected

- 5) Line 806 – Was the commercial adjuvant AS01b used or was an analogue used? I don't see mention of a GSK collaboration.

The AS01b adjuvant used was the original GSK adjuvant, that was obtained as the adjuvant that accompanies the commercially available Shingrix vaccine. This information is added to the revised version of the manuscript.

REVIEWERS' COMMENTS

Reviewer #1 (Remarks to the Author):

The authors have responded robustly to reviewer comments.

Other than claims of priority (e.g. Abstract - "...showed for the first time...") the revised paper is fine.

Reviewer #2 (Remarks to the Author):

My concerns have been addressed.